# A new conceptual framework for assessing physical state of the Baltic Sea

Urmas Raudsepp[1], Ilja Maljutenko[1], Priidik Lagemaa[1], Karina von Schuckmann[2]

[1] Department of Marine Systems, Tallinn University of Technology, Tallinn, 12618, Estonia

[2] Mercator Ocean international, 2 Av. de l'Aérodrome de Montaudran, 31400 Toulouse

*Correspondence to*: Ilja Maljutenko (ilja.maljutenko@taltech.ee)

**Abstract.**

Climate change is placing growing pressure on all parts of the ocean, increasing the need for regular information to support regional assessments and inform policy and decision-making. Understanding not only what is changing and where, but also why, is essential for effective response and meaningful action. To answer this need a new conceptual framework for the assessment of the physical state of the general natural water basin was introduced and then tested for the Baltic Sea. The approach is based on major process characteristics of the Baltic Sea and includes the analysis of mutual variability of well established climate indicators such as ocean heat content (OHC), freshwater content (FWC), subsurface temperature and salinity, and combined with atmospheric forcing functions along with salt transport across the open boundaries as well as river runoff. A random forest model is used as the main analyses tool to enable statistical dependencies between state variables and potential forcing factors. Results reveal a clear 30-year warming trend in the Baltic Sea, closely linked on an interannual scale to 2-meter air temperature, evaporation, and wind stress magnitude. The study highlights that interannual variations in temperature and salinity within the vertically extended halocline are key drivers of changes in OHC and FWC in the Baltic Sea. Interannual changes of FWC are explained by large volume saline water inflows, net precipitation and zonal wind stress. This framework also offers a new perspective of the potential impact of a shallowing mixed layer depth, resulting from sustained sensible heat flux changes at the air-sea interface, on salt export and the overall reduction of FWC in the Baltic Sea. This new framework could be applied to other geographical regions or future datasets, providing consistent information for a basin-wide monitoring tool that tracks the state and variability of the sea. Such a tool could be integrated into regional climate and environmental assessments.

**Short Summary.** In the last three decades, the Baltic Sea has experienced an increase in temperature and salinity. This trend aligns with the broader pattern of atmospheric warming. The significant warming and the yearly fluctuations in the ocean's heat content in the Baltic Sea are largely explained by subsurface temperature variations in the upper 100-meter layer, which includes the seasonal thermocline and the permanent halocline. These fluctuations are influenced by factors such as air temperature, evaporation, and the magnitude of wind stress. The changes in the sea's liquid freshwater content are primarily driven by salinity shifts within the halocline layer, which extends vertically from 40 to 120 meters depth. However, salinity changes in the upper layer play a minor role in the yearly variability of the freshwater content. The inflow of saline water,

overall precipitation, and zonal wind stress are the principal factors affecting the freshwater content changes in the Baltic Sea.

## 1 Introduction

Human-induced greenhouse gas emissions are warming Earth's climate, causing ocean temperatures to rise and ice to melt globally (IPCC, 2021). The increase in ocean water temperatures has induced a rise in Ocean Heat Content (OHC), and ice melt on land has introduced significant amounts of freshwater into the ocean, contributing to the rise in global sea levels. In 2023, global average sea surface temperature reached a record high relative to the 1973–2024 baseline period (McGrath et al., 2024), and global ocean heat content climbed to record levels (Cheng et al., 2024). In the Baltic Sea, the temperature trends for the period 1850-2008 show fast warming at the surface ($\sim 0.06$ K decade$^{-1}$) and bottom ($> 0.04$ K decade$^{-1}$), and slow in the intermediate layers ($< 0.04$ K decade$^{-1}$) (Dutheil et al., 2023). Surface warming has progressively increased over time, primarily due to the sensible heat flux and latent heat flux (Kniebusch et al., 2019a). Trends in Fresh Water Content (FWC) are not as consistent globally as those of OHC (Boyer et al., 2007), although the rise in global sea level is widely acknowledged (Frederikse et al., 2020). Salinity patterns differ across various ocean regions of the world (Skliris et al., 2014), with the North Atlantic–North Pacific salinity contrast increasing by 5.9% ± 0.6% since 1965 (Lu et al., 2024). At a regional scale in the Baltic Sea, FWC has shown a significant downward trend over the last 30 years (Raudsepp et al., 2023). Winsor et al. (2001) highlighted the cumulative impact of riverine input on the Baltic's freshwater budget, while Rodhe and Winsor (2002) underscored the importance of episodic saltwater inflows in renewing deep water. An increase in freshwater supply to the Baltic Sea will intensify the regional water cycling, resulting in lower salinity, and vice versa.

The analysis of the physical state of natural water basins typically focuses on the evolution and spatial distribution of temperature and salinity and corresponding uncertainty estimations , which are essential ocean variables (EOV, Lindestroem et al. 2012). These variables are four dimensional and therefore provide spatially and temporally resolved description of the state of the water body. Meanwhile, OHC and FWC are vital integral characteristics of the ocean, indicative of a water body's energy and mass, respectively. OHC offers a comprehensive view of oceanic heat storage, crucial for evaluating climate change impacts, energy budgets, and long-term trends (Forster et al., 2024). FWC represents the mass of the freshwater relative to the total mass of a water parcel with a given salinity (see Raudsepp et al., 2023). The increase of net precipitation over land and sea areas, decrease of the ice cover and increase of river runoff are the main components of the global hydrological cycle that increase FWC in the ocean (Boyer et al., 2007; Cheng et al., 2020; Yu et al., 2020). While OHC is a well-established indicator in ocean and climate research, its counterpart, ocean FWC, has received less attention.

We propose a new conceptual framework for assessing the physical state of the Baltic Sea by integrating multiple physical and statistical approaches (Fig. 1). OHC and FWC serve as integrative indicators of the Baltic Sea's physical state, analogous to essential climate indicators (IPCC, 2021; Forster et al., 2025). The OHC and FWC are well-established measures (IPCC,

2021; Forster et al., 2025), which we integrate into a unified assessment framework with additional analysis layers - vertical distribution and statistical inference to assess the Baltic Sea's state and are central to understanding its energy and mass balance. OHC reflects the vertically integrated heat stored in the water column and is primarily influenced by surface heat fluxes, vertical mixing, and subsurface temperature changes (Forster et al., 2025). FWC quantifies the deviation of the water column's salinity from a reference value and serves as a measure of accumulated freshwater (Durack, 2015; Raudsepp et al., 2023). It is affected by net precipitation, river runoff, evaporation, and saltwater intrusions from the North Sea. In this study, these indicators are integrated into a unified assessment framework that includes both their vertical structure and statistical inference layers. The study identifies the importance of these major variables affecting the OHC and FWC , including subsurface temperature, salinity, atmospheric forcing factors, and salt transport.

The framework follows a three-stage process: time-series analysis, depth-based variability analysis and statistical relationships using machine learning. The initial phase consists of calculating the time series of OHC and FWC for the entire Baltic Sea. This provides insights into long-term trends and interannual variability. In basins covered partially by sea ice, the annual mean ice extent (MIE) is considered an important integral characteristic. The next step examines the horizontally averaged vertical distribution of temperature (for OHC) and salinity (for FWC) to determine which depth ranges contribute the most to their variations. While this does not directly attribute causal links, the vertical profiles of temperature and salinity provide strong indications of which forcing factors might be responsible for changes in OHC and FWC. The final stage integrates forcing functions and ocean state characteristics to identify statistical dependencies between them, using a Random Forest (RF) model to probe potential drivers of variability. A RF model is employed to highlight statistical dependencies between oceanic state variables and external forcing mechanisms. This machine-learning approach enables the identification of general patterns in the temporal evolution of the Baltic Sea's physical state. The main reason we introduced the RF models is to determine, in a data-driven way, the relative importance of different depth layers and forcing factors on the variability of OHC and FWC. The RF approach offers a flexible means to handle non-linear relationships and multiple predictors simultaneously.

Our proposed framework integrates the analysis of OHC and FWC by considering both their bulk integral values and their vertical distributions, allowing for the identification of key depth ranges contributing to their variability – which goes beyond other similar frameworks. Unlike the GOOS EOV framework (https://goosocean.org/), which focuses on structured global ocean monitoring without machine learning-based statistical analysis, our approach explicitly incorporates machine learning to identify potential drivers of variability. Compared to the IPCC Climate and Ocean Monitoring Framework (IPCC AR6 (2021) Ocean Observations Chapter https://www.ipcc.ch/report/ar6/wg1/), which relies on dynamical climate models for global-scale processes, our framework is designed for regional-scale Baltic Sea analysis, offering a more localized and detailed assessment. Finally, while the NASA Salinity and Heat Budget Analysis (NASA Salinity Budget Project https://podaac.jpl.nasa.gov) is largely empirical and focused on global salinity and heat transport, our approach provides a structured three-stage methodology, incorporating not only empirical analysis but also a cause-and-effect exploration using

machine learning. This makes our framework uniquely suited for regional climate monitoring and actionable insights into the physical state of the Baltic Sea.

The Baltic Sea is recognized for its spatially pronounced heterogeneous structure. Its various subregions may exhibit distinct temporal variations in key state variables and overall dynamics, making it a complex environment for testing the conceptual framework. The Baltic Sea, a shallow marginal sea in northeastern Europe, is characterized by its hydrographic fields and sea ice conditions (Leppäranta and Myrberg, 2009). Salinity levels are affected by saline water inflows from the North Sea through the Danish straits, riverine freshwater inputs, and net precipitation (Lehmann et al., 2022). Major Baltic Inflows, which introduce saline and oxygen-rich water, are sporadic and unpredictable (Mohrholz, 2018). Temperature fields are influenced by the heat exchange with the atmosphere. The residence time of the Baltic Sea's water is several decades long (Meier et al., 2022). The vertical salinity stratification is defined by the halocline's depth, featuring a well-mixed surface layer and a slightly stratified layer beneath. Water temperature plays a crucial role in forming secondary stratification related to the temperature of the upper mixed layer. Seasonal temperature cycles lead to partial freezing of the Baltic Sea in winter. Changes in sea ice extent over time are a vital indicator of climate change for the area. A reduction in maximum ice extent impacts the sea's vertical stratification and the seasonal trends in ocean heat and freshwater content (Raudsepp et al., 2022; 2023). Despite global warming, there has not been a significant increase in the Baltic Sea's relative sea level (Ranasinghe et al., 2021), which instead shows a strong seasonal cycle.

This conceptual framework is designed as an indicator-based approach relevant to policymakers. OHC and FWC distill complex, high-dimensional data (many temperature and salinity profiles) into two easy-to-interpret indices of the Baltic Sea's thermal and haline state. This kind of simplification is valuable for decision-makers who require clear, high-level indicators. However, interpretation is also necessary—and this becomes particularly challenging at the regional scale, where a variety of interacting processes, including long-term changes, are at play. The framework not only delivers time series and regular statistical assessments, but also provides a structured path toward meaningful interpretation by focusing directly on the main drivers of change. Understanding not just what is changing and where, but also why it is happening, is essential for taking informed action and gaining a comprehensive view of the system. The framework enables the monitoring of climate change impacts on the Baltic Sea while maintaining a balance between scientific rigor and practical accessibility. It is not meant to serve as a comprehensive dynamical model but rather as a tool for assessing the state of the Baltic Sea and guiding regional management decisions. The framework is grounded in well-established physical quantities and validated by statistical analysis, which ensures that its findings are consistent and credible.

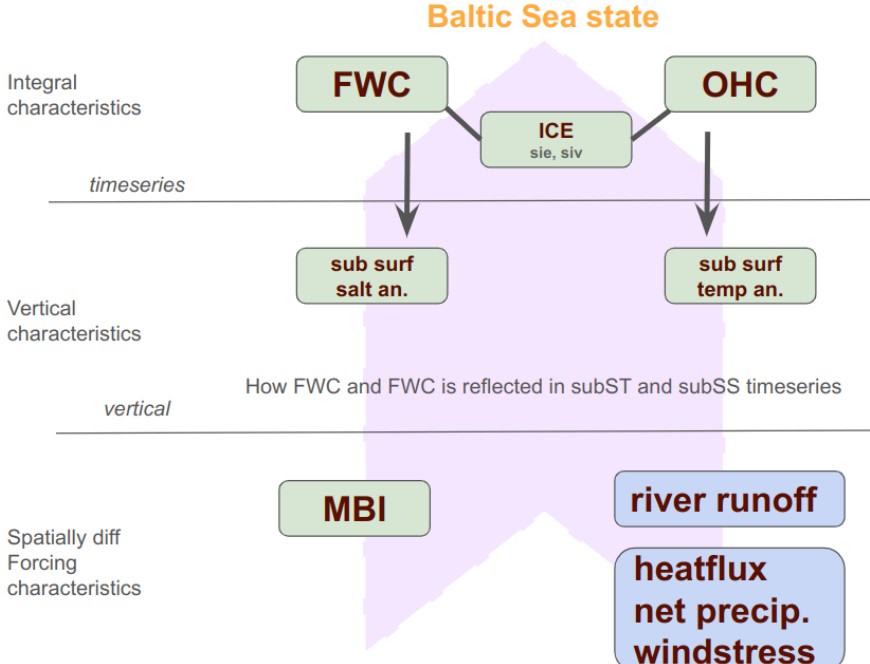

**Figure 1**: Conceptual Scheme of the Baltic Sea State parameters illustrating the interplay among key indicators: ocean heat content (OHC), freshwater content (FWC), sea ice extent (SIE), sea ice volume (SIV), subsurface temperature (subST), subsurface salinity (subSS), and major Baltic inflows (MBI). Changes in OHC and FWC drive variations in sea ice cover and subsurface conditions, while episodic MBI events inject saline water into deep layers, altering subsurface salinity and temperature. Together, these processes shape the overall state of the Baltic Sea.

The study aims to present a framework for assessing the physical state of the Baltic Sea by integrating annual mean values of OHC, FWC, subsurface temperature and salinity, atmospheric forcing functions, salt transport, and river runoff. The objective is to use a data-driven RF approach as the primary analysis tool to parse out nonlinear relationships and feature importances from a broad dataset. This study introduces an integrative, basin-wide approach, defining the entire Baltic Sea as a single water body for analysis. It computes a time series of total OHC and FWC for the whole sea. Unlike previous approaches that focus mainly on local variations, this methodology prioritizes integrated indices that capture the sea's overall state. This holistic perspective represents a fundamental shift away from fragmented, localized analyses toward a comprehensive understanding of ocean dynamics, making the framework uniquely suited to inform large-scale assessments and decision-making.

## 2 Data and methods

**Table 1**: Product Table

| Product ref. no. | Product ID & type | Data access | Documentation |
|---|---|---|---|
| 1 | BALTICSEA_MULTIYEAR_PHY_003_011; Numerical models | EU Copernicus Marine Service Product (2023); | Quality Information Document (QUID): Panteleit et al. (2023); Product User Manual (PUM): Ringgaard et al. (2024) |
| 2 | ERA5; Numerical models | Copernicus Climate Change Service (2023) | Product reference: Hersbach et al., 2023 Journal article: Hersbach et al., 2020 |
| 3 | E-HYPE; Numerical models | SMHI | Donnelly et al., 2016 |

## 2.1 Oceanographic and atmospheric data

The Baltic Sea physics reanalysis multi-year product (BAL-MYP; Table 1 product ref. no. 1) is derived from the ocean model NEMO v4.0 (Gurvan et al., 2019). It assimilates satellite observations of sea surface temperature (SST) (EU Copernicus Marine Service Product, 2022) and in-situ temperature and salinity profiles from the ICES database (ICES Bottle and low-resolution CTD dataset, 2022). The model data is provided on a grid with a horizontal resolution of 1 nautical mile, including 56 vertical layers, covering the entire Baltic Sea and the transition zone to the North Sea. The dataset covers the period from 1993 to 2023, with the model setup detailed in the Product User Manual (PUM, Ringgaard et al., 2024).

The BAL-MYP has been extensively validated, as documented in the Quality Information Document (QUID; Panteleit et al., 2023), focusing on the period from 1st January 1993 to 31st December 2018. Additionally, the BAL-MYP data were evaluated using a clustering method with the K-means algorithm (Raudsepp and Maljutenko, 2022), which provided insights into the reanalysis accuracy by categorising errors (Lindenthal et al., 2023). Fifty-seven percent of the data are clustered with a bias of dS=-0.40 g/kg and dT=-0.02 °C, encompassing 57% of all data points with RMSE S=0.92 g/kg and T=0.54 °C. These points are distributed throughout the Baltic Sea. Clusters with high positive and negative temperature biases account for 11% and 8% of total points, respectively, with marginal salinity biases and relatively even spatial distributions across the Baltic Sea. Twenty-six percent of the points have low temperature but high salinity errors, both negative and positive, predominantly located in the southwestern Baltic Sea, indicating occasional underestimation or overestimation of the inflow/outflow salinity.

Given its spatial coverage and validated accuracy, the BAL-MYP reanalysis (Table 1, product ref. no. 1) provides a reliable basis for calculating integrated environmental indicators such as OHC and FWC, which are essential for large-scale climate assessments. OHC directly reflects Earth's energy imbalance, making it a key metric for tracking global warming, unlike basin-averaged temperature, which lacks a direct connection to energy budgets (von Schuckmann et al., 2016, 2023). Consequently, OHC is prioritized in climate models and international assessments (IPCC, 2019) due to its direct relationship

with anthropogenic forcing and its predictive value for future climate scenarios. The daily OHC has been computed for each model grid cell from reanalysis (product ref. no. 1), following the methodology of Meyssignac et al. (2019)

$$OHC = \rho * cp * ( T +273.15) \qquad (1),$$

where $\rho$ is the density of seawater calculated following the TEOS10 (IOC et al. 2010), cp is specific heat capacity calculated as a third order polynomial function of salinity and temperature according to Millero et al.(1973), T is daily temperature.

Ocean FWC is deemed more significant than mean salinity for understanding climate dynamics and ocean processes. FWC provides a holistic measure of freshwater storage and its effects on ocean circulation, climate, and sea-level rise (Solomon et al., 2021; Fukumori et al., 2021). It directly measures freshwater inputs (e.g., ice melt, river runoff, rainfall) or losses (e.g., evaporation), whereas mean salinity only indicates the average salt concentration, ignoring volume (Hoffman et al., 2023). A minor salinity change over a large water volume could signify a substantial freshwater flux, which mean salinity alone would not reveal (Schauer and Losch, 2019). The FWC was calculated at each grid point and day as Boyer et al. (2007)

$$FWC = \rho(Sref, Tref, p) / \rho(0, Tref, p ) \cdot ( Sref - S) / S \qquad (2)$$

The three-dimensional temperature (Tref) and salinity (Sref) fields are temporal averages over the period of 1993–2023. A detailed description of the calculation procedure is available in Raudsepp et al. (2023). The OHC and FWC were calculated by spatially integrating the gridded OHC, (1), and FWC, (2), over the Baltic Sea, and then the annual mean OHC and FWC values were calculated from these daily values.

The Mixed Layer Depth (MLD), also referred to as the Upper Mixed Layer (UML), was included in the analysis using data from a multi-year reanalysis product (product ref. no. 1). The MLD was calculated based on density stratification following the method of de Boyer Montégut et al. (2004), which defines MLD as the depth at which seawater density deviates from the reference density at 10 m depth by a specified threshold. For the Baltic Sea, this threshold was adjusted to 0.03 kg/m³ to better represent the characteristics of the regional upper mixed layer (Panteleit et al., 2023).

Atmospheric data for the RF input (Atm8) were obtained from the ERA5 reanalysis (product ref. no. 2) for the period 1993–2023. The parameters (8 in total) included 2-meter air temperature, total precipitation, evaporation, wind stress magnitude, and the x- and y-components of wind stress, along with total cloud cover and surface net solar radiation. The time series for the annual mean values of these atmospheric parameters were computed as horizontal averages across the Baltic Sea region (8 °E - 33 °E and 52 °N - 68 °N).

Additionally, total river runoff to the Baltic Sea (RNF) (product ref. no. 3) and a proxy for saltwater inflows — represented by bottom salinity in the Bornholm Basin (SOB) (product ref. no. 1) — were included as external forcing factors. These variables capture key hydrological and oceanographic influences not fully accounted for by atmospheric drivers alone, and contribute to a more comprehensive assessment of interannual variability in FWC.

Horizontally average temperature and salinity profiles calculated from the BAL-MYP (product ref. no. 1) at 42 different depth layers (shown on Fig. 3) and Baltic Sea domain (13 °E - 31 °E and 53 °N - 66 °N; excluding the Skagerrak strait) were used as predictors in two of the RF models. The rationale for using the full vertical profiles is to allow the model to identify which depth layers most strongly influence the total OHC or FWC. Instead of assuming a priori which depths matter, the RF can learn this from data: if variations at a particular depth are consistently associated with changes in total OHC/FWC, the model's feature importance for that depth will be high.

## 2.2 Random Forest

Random Forest (RF) is an ensemble learning method predominantly used for classification and regression tasks (Breiman, 2001). It functions by building multiple decision trees during the training phase and outputs the class that is the mode of the classes (classification) or the mean prediction (regression) of the individual trees. This method enhances accuracy and helps prevent overfitting, thus making it resilient to noise in the dataset. RF proves to be highly effective in analyzing complex interactions between variables, such as the relationships between marine state variables and atmospheric parameters. Its effectiveness is due to its capability to manage high-dimensional data and its resistance to outliers and noise, which are prevalent in environmental datasets. Additionally, RF is adept at detecting nonlinear relationships between predictor variables (atmospheric parameters) and response variables (marine state variables), which linear models often overlook.

In the context of an RF model, feature importance is a technique that identifies the most influential input features (variables) in predicting the output variable. The importance of each feature is determined by the decrease in model accuracy when the data for that feature is permuted, while all other features remain unchanged. If permuting a feature's values significantly increases the model's error, that feature is deemed crucial for the model's predictions. This approach aids in discerning the contribution of each feature to the model's decision-making process and in identifying key atmospheric parameters that significantly impact marine state variables. A positive value for a feature implies that permuting that predictor variable's values raises the model's prediction error, indicating the variable's importance for the model's predictive accuracy. A higher positive value suggests greater reliance on that variable by the model.

In this study we have trained the four different RF models to fit the OHC and FWC annual average timeseries from annual average predictor variables with the hyperparameter configurations shown in Table 2. Two models are trained to predict the OHC and FWC values from the set of the atmospheric variables (VAR arguments). The OHC model uses only atmospheric input variables, whereas the FWC model includes, in addition to atmospheric variables, two external predictors: total river runoff to the Baltic Sea and bottom salinity in the Bornholm Basin. In addition, two more models are trained to predict OHC and FWC using horizontally averaged temperature and salinity profiles (Z arguments). To study variability independent of long-term trends, all input variables and target time series used in the VAR models were linearly detrended prior to training. This ensures the models capture interannual to decadal fluctuations rather than long-term changes.

To optimize the performance of the RF models while ensuring robustness and generalizability, a set of hyperparameters was
selected based on best practices outlined by Probst et al. (2019), along with and based on sensitivity analysis conducted for
the number of trees (Fig A2). The minimum leaf size (MinLS) was set to 1, allowing the trees to fully grow and capture
complex data patterns. The number of predictors to sample at each split (Pred2Samp) was dynamically determined as
one-third of the total number of predictors, tackling a balance between feature randomness and predictive strength. This
approach promotes diversity among trees while preventing excessive correlation. The number of trees (NumTrees) in each
RF model was set to 100, providing sufficient ensemble stability while maintaining computational efficiency (Appendix 2).
Since this study employs RF models to investigate nonlinear relationships between predictors and state variables, we use the
entire dataset (all available data) as the training set to maximize the models' ability to learn patterns. We conducted 5-fold
cross-validation, which yielded similar conclusions regarding which predictors are most influential, suggesting that the RF
importance measures are qualitatively robust. To further enhance predictive reliability, assess uncertainty, and evaluate the
stability of both predictions and feature importances, an ensemble of 150 independently trained RF models was constructed.

We employed MATLAB's TreeBagger function to assess the feature importance of atmospheric predictors on marine state
variables. The `OOBPermutedPredictorDeltaError` method, a robust metric from MATLAB's TreeBagger, quantifies each
predictor's importance via the out-of-bag (OOB) prediction error. This involves permuting each variable's values across OOB
observations for each tree. The resulting change in prediction error from these permutations is calculated for each tree. These
measures are averaged across all trees and normalised by the standard deviation of the changes, providing a standardised
score that highlights the variables with the most significant impact on predictive accuracy. Averaging the feature importance
scores across all models in ensembles minimises the noise and variability from any single model's training, offering a more
consistent and dependable indication of each atmospheric parameter's contribution to predicting marine state variables. A
larger importance value means that permuting (randomizing) that predictor greatly degrades model accuracy, indicating the
predictor was influential. Conversely, near-zero or negative importance means that randomizing the predictor had little effect
or even slightly improved the model's error, suggesting the predictor is not informative (or that its influence is redundant or
noisy).

**Table 2.** Hyperparameter configurations and validation for different Random forest models. All models use the same
Random Forest configuration: number of trees set to 100 and forest ensemble size to 150. The variable number of predictors
to sample at each split (Pred2Samp) is set ⅔ of the number of input parameters. The minimum leaf size is fixed at 1.
Asterisks (*) indicate RF models applied to variability using detrended VARiables. Models performance is shown by means
of pearson correlation coefficient (CC) and root mean square difference (RMSD).

| Model | Predictors | Pred2Samp | CC | RMSD |
|---|---|---|---|---|
| RF_OHC(Z) | Tprof_42[1] | 14 | 0.986 | 0.0016 |
| RF_FWC(Z) | Sprof_42[1] | 14 | 0.973 | 0.004 |

| | | | | |
|---|---|---|---|---|
| RF_OHC(VAR)* | ATM_8[2] | 3 | 0.9012 | 0.3432 |
| RF_FWC(VAR)* | ATM_8[2]+RNF[3]+SOB[4] | 4 | 0.8994 | 0.3624 |

[1]Tprof_42, Sprof_42: Horizontally averaged annual mean temperature and salinity profiles at 42 depth levels (Fig 3).

[2]ATM_8: Horizontally averaged annual mean values of eight atmospheric variables.

[3]RNF: Total annual river runoff into the Baltic Sea.

[4]SOB: Annual mean bottom salinity in the Bornholm Basin.

**3 Results**

Both OHC and FWC display a statistically significant linear trend, as shown in Figure 2. Using a z-score time series allows
for the comparison of trends per year (trend*) and data distributions without the influence of their units. OHC shows an
increasing trend* of 0.089±0.025, while FWC exhibits a decreasing trend* of -0.092±0.023, both comparable in magnitude
(Table 3). The corresponding absolute values are 0.34±0.095 W/m$^2$ for OHC and -36.99±9.20 km$^3$/year for FWC (Table 3).
Between 1993 and 2003, OHC and FWC varied similarly, both rising and falling concurrently (blue dots in Fig. 2). After this
period, their patterns diverged (yellow and red dots in Fig. 2). Interannual variations of the annual mean sea ice extent and
OHC are strongly correlated but in opposite phases (not shown). Among the forcing functions, the 2-meter air temperature
shows a distinct positive trend (Fig. 2), albeit weaker than the trends of OHC and FWC (Table 3). The air temperature over
the Baltic Sea area has risen with trend* of 0.074±0.031 (Table 3). Surface net solar radiation has a weaker but still
significant positive trend* of 0.058±0.035, and the evaporation time series shows a negative trend* of -0.041±0.039 (Fig. 2,
Table 3). Other atmospheric variables did not exhibit statistically significant trends (Fig. 2). Correlation coefficients among
various atmospheric datasets were generally low (Table 4). The two highest correlation coefficients, 0.76 and 0.73, are
between wind stress magnitude and its zonal component, indicating a predominance of westerly airflow over the Baltic Sea,
and between 2-meter air temperature and surface net solar radiation, respectively. The low correlations suggest a weak
statistical relationship between the annual mean atmospheric parameters, supporting the inclusion of all forcing functions in
the RF model.

**Table 3.** Linear annual trend values of z-scored time series (trend*), standard deviation (STD), linear trend of physical value
(Unit/year, except for OHC) and mean value (mean) of original time series. *OHC*: ocean heat content, *FWC*: fresh water
content, *T2*: 2 metre temperature, *TP*: total precipitation, *EVAP*: evaporation, *Wstr*: windstress, *WUstr*: windstress u
component, *WVstr*: windstress v component, *TCC*: total cloud cover , *SSR*: surface net solar radiation, *RNF*: river runoff.

| Variable: | OHC | FWC | T2 | TP | EVAP | Wstr | WUstr | WVstr | TCC | SSR | RNF |
|---|---|---|---|---|---|---|---|---|---|---|---|
| Unit | MJ/m² | km³ | °C | m/y | m/y | N/m² | N/m² | N/m² | 1 | W/m² | m³/s |
| trend*: | 0.089 ± 0.025 | -0.092 ± 0.023 | 0.074 ± 0.031 | 0.032 ± 0.04 | -0.041 ± 0.039 | -0.0016 ± 0.0418 | 0.013 ± 0.041 | 0.015 ± 0.041 | -0.0077 ± 0.0417 | 0.058 ± 0.035 | 0.0073 ± 0.0417 |

| | | | | | | | | | | | |
|---|---|---|---|---|---|---|---|---|---|---|---|
| STD: | 122.02 | 402.00 | 0.73 | 0.071 | 0.041 | 0.0056 | 0.0100 | 0.0072 | 0.0226 | 3.16 | 1,687.92 |
| trend: | 0.344 (W/m²) | -36.987 | 0.054 | 0.0023 | -0.0016 | $-8.85 \times 10^{-6}$ | $1.32 \times 10^{-4}$ | $1.05 \times 10^{-4}$ | $-1.75 \times 10^{-4}$ | 0.18 | 12.31 |
| mean: | 60.20 | -63.73 | 7.65 | 0.73 | -0.55 | 0.0999 | 0.0244 | 0.0138 | 0.6493 | 113.92 | 17,807.77 |

**Table 4.** Correlations coefficients (lower triangle) and StandardErrors (Gnambs, 2023) (upper triangle) of atmospheric parameters. Correlation coefficients which pass two-tailed t-test at 95% confidence are in bold. *OHC*: ocean heat content, *FWC*: fresh water content, *T2*: 2 metre temperature, *TP*: total precipitation, *EVAP*: evaporation, *Wstr*: wind stress magnitude, *WUstr*: wind stress u component, *WVstr*: wind stress v component, *TCC*: total cloud cover , *SSR*: surface net solar radiation.

| | *T2* | *TP* | *EVAP* | *Wstr* | *WUstr* | *WVstr* | *TCC* | *SSR* |
|---|---|---|---|---|---|---|---|---|
| *T2* | | 0.19 | 0.17 | 0.17 | 0.15 | 0.14 | 0.15 | 0.09 |
| *TP* | 0.12 | | 0.18 | 0.17 | 0.18 | 0.18 | 0.13 | 0.17 |
| *EVAP* | -0.28 | -0.18 | | 0.19 | 0.18 | 0.16 | 0.19 | 0.15 |
| *Wstr* | 0.31 | 0.35 | -0.10 | | 0.08 | 0.15 | 0.18 | 0.19 |
| *WUstr* | **0.47** | 0.25 | 0.16 | **0.76** | | 0.15 | 0.16 | 0.18 |
| *WVstr* | **0.48** | 0.16 | 0.37 | **0.43** | **0.43** | | 0.19 | 0.19 |
| *TCC* | -0.43 | **0.58** | -0.04 | -0.20 | -0.42 | -0.13 | | 0.09 |
| *SSR* | **0.73** | -0.31 | -0.43 | 0.07 | 0.18 | 0.11 | -0.73 | |

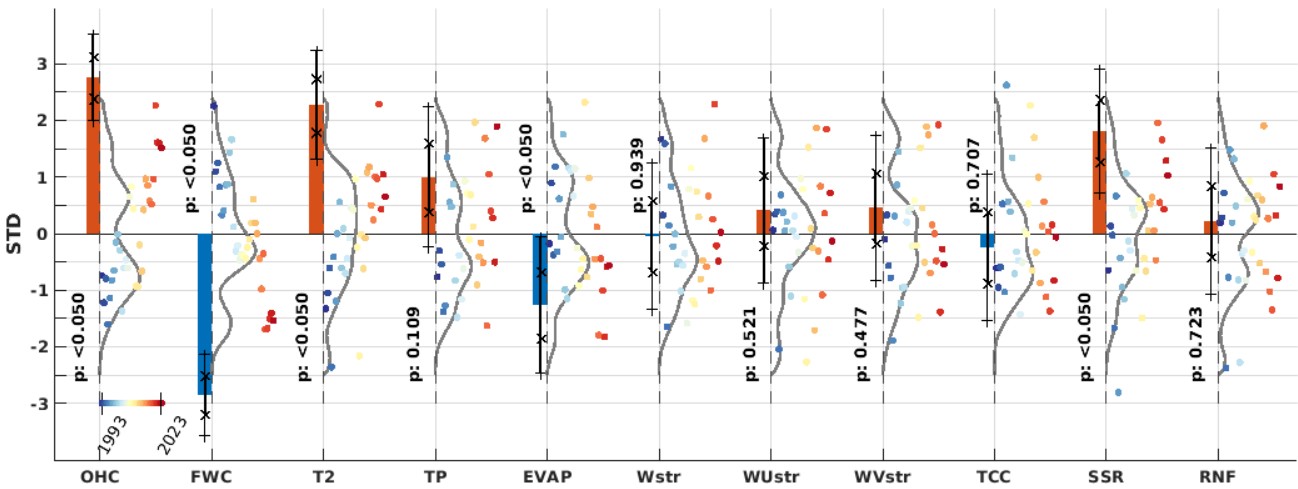

**Figure 2**: Trend analysis and probability distribution functions (PDFs) of the annual time series of standardized (*z-scores) Baltic Sea state and meteorological parameters. To the left of the dashed line, the period-normalized annual trend values (multiplied by the period length in years i.e. 30) are displayed as red (positive) and blue (negative) bars with corresponding p-values (95% confidence level), along with whiskers representing ±1 standard error (x ticks) and the 95% uncertainty range (+ ticks). On the right side from the dashed line, probability density functions (PDFs) are

288 shown as the solid lines for the standardized time series, which are represented by colored dots. The color of the dots

represents the year on a common color scale shown at the OHC variable.

For each dashed axis following variable stands *OHC*: ocean heat content, *FWC*: fresh water content, *T2*: 2 metre

temperature, *TP*: total precipitation, *EVAP*: evaporation, *Wstr*: windstress, *WU/WVstr*,: windstress u and v component,

*TCC*: total cloud cover , *SSR*: surface net solar radiation, *RNF*: river runoff.

In analyzing OHC variations, we use a RF_OHC(Z) model (Table 2). This model employs horizontally averaged annual

temperature values at each depth level, derived from the depth levels of a multi-year product (product ref. no. 1), as input

features. The RF model finely replicates the annual OHC time series (Fig 3a), with high correlation coefficient (0.986) and a

RMSD of the standardized time series at 0.0016. However, it did not capture the extreme OHC event in 2020 or the low

OHC extreme in 1996 (Fig. 3). Feature importance is significant within a depth range of 10-80 meters (Fig. 3b), with two

peaks at depths of 18 and 60 meters, aligning with the average depths of the seasonal thermocline and the permanent

halocline, respectively. This suggests that interannual OHC variations are mainly influenced by temperature changes within

these layers. Subsurface temperatures from 1993 to 2023 indicate warming trends of approximately 0.06 °C/year across all

depths (CMS 2024a). From 1993 to 1997, deep water temperatures remained relatively low (below 6 °C). Since 1998, deeper

waters have warmed, with temperatures above 7 °C occupying the layer below 100 meters since 2019. The water

temperature below the halocline has risen by about 2 °C since 1993, and the cold intermediate layer's temperature has also

increased during the 1993-2023 period.

A similar method is employed to elucidate the inter-annual fluctuations of FWC using RF_FWC(Z) (Table 2), utilizing

horizontally averaged salinity at each depth level. The model's precision is slightly lower (Correlation: 0.973, RMSD of

standardized time series: 0.004) compared to that for OHC. The model consistently underperforms in predicting the FWC

peaks, encompassing both the lows and highs (Fig. 3c). The most notable features cover the depth range of 40-120 meters

(Fig. 3d), coinciding with a halocline layer and its vertical extensions to both shallower and deeper depth. The salinity levels

at the bottom layer are of secondary importance to the inter-annual variations of FWC in the Baltic Sea. The salinity in the

top 25-meter stratum exerts a minimal influence on FWC changes. The interannual variability of salinity in the upper stratum

is minor relative to the deeper stratum. The salinity gradient ascends steadily from zero at a depth of 25 meters to 0.04 g/kg

annually at 70 meters (CMS 2024b). The most marked trend, 0.045 g/kg per annum, occurs within the expanded halocline

layer extending from 70 to 150 meters. Notably, there is a slight dip in the salinity trend to 0.04 g/kg per annum between the

depths of 150 and 220 meters. While this reduction is slight, it indicates that salt influx into the expanded halocline layer is

more significant than into the deeper strata. A salinity trend of 0.05 g/kg annually is detected in the deepest stratum of the

Baltic Sea.

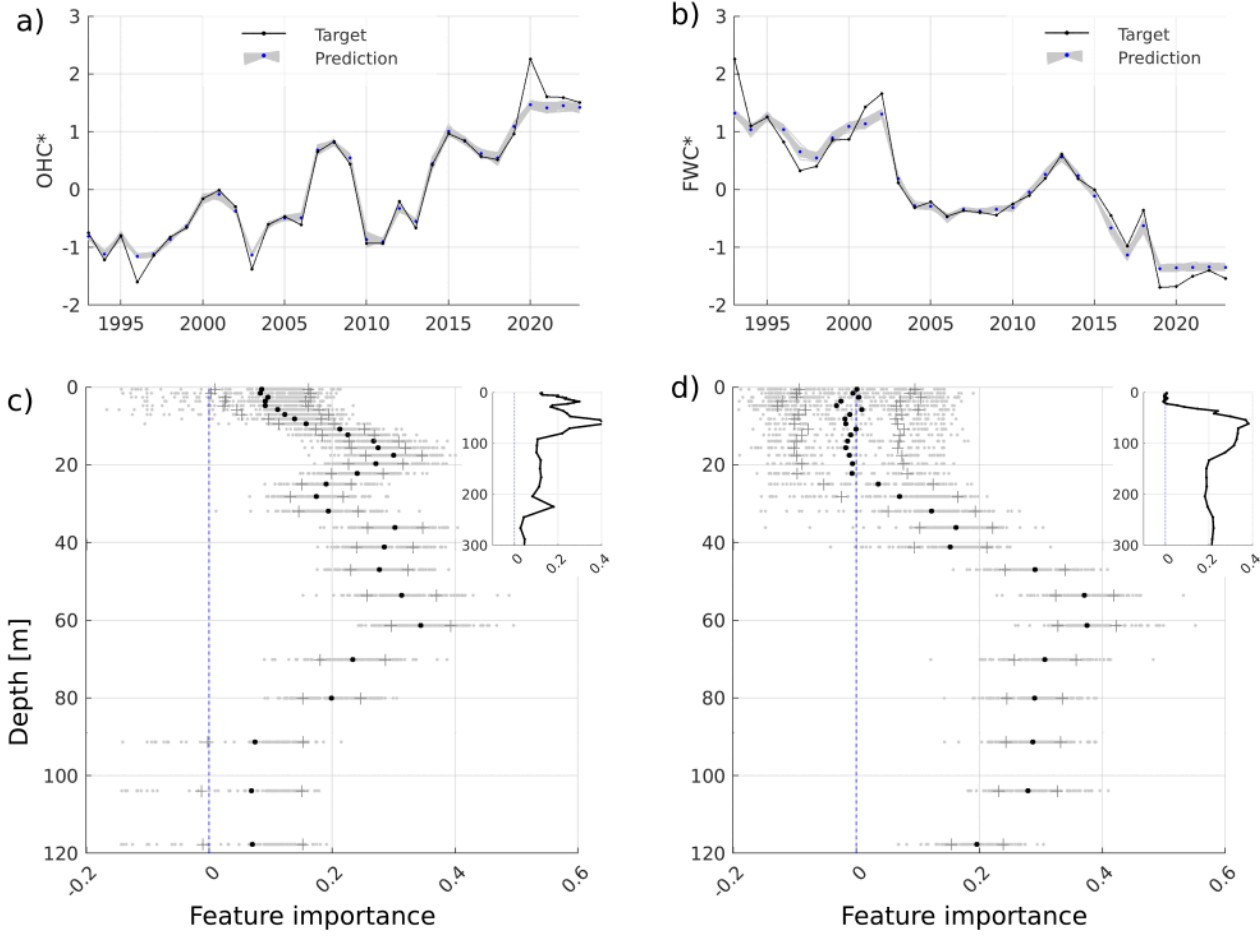

**Figure 3**: OHC* and FWC* ensemble predictions (ens. mean as blue dots) using the horizontal average salinity and
temperature profiles (a), (b). The prediction features importance, with ensemble spread (1 STD shown with "+" marker),
for each depth in the upper 120 m layer shown on c) and d) and for full depth range in the upper-right inset panels. All
variables are z-scored.

Building a RF model targeting OHC and FWC timeseries with atmospheric forcing functions reveals the 2-meter air

temperature as the most significant contributor (Appendix 1). This correlation is physically plausible for OHC but less so for

FWC. The 2-meter air temperature affects the air-sea heat exchange via the sensible heat flux component. To further explore

the declining FWC trend, we examined interannual changes in the annual average upper mixed layer depth (MLD). In the

Baltic Sea, MLD varies widely across different areas and seasons. A shallowing of MLD is observed in the Baltic Proper and

to some extent in the Bothnian Sea, while a MLD deepening is noted in the Bothnian Bay, the Gulf of Finland, and the Gulf

of Riga. Typically, the Baltic Sea's stratification is influenced by salinity, although a seasonal thermocline forms across the

sea. In the northern and eastern basins, the dispersal of river water during spring and summer leads to the development of the

seasonal pycnocline. Conversely, in the southern Baltic Sea, the spread of river water is mostly restricted to the coastal areas, so the mixed layer is less affected by the seasonal halocline.

We performed test experiments with the RF model, incorporating the upper mixed layer (UML) as an additional feature. We determined the annual mean UML depth across the Baltic Sea and specifically for the Eastern Gotland Basin. The decline in the UML depth was more significant in the Eastern Gotland Basin compared to the entire Baltic Sea. The UML depth in the Eastern Gotland Basin decreased from 30 meters in 1993 to 22 meters in 2023. The MLD feature became more significant than the 2-meter temperature in explaining the FWC when we considered the UML depth in the Eastern Gotland Basin. However, the results were contentious when we applied the average UML depth for the entire Baltic Sea. An increase in the 2-meter temperature may cause a shallower mixed layer, potentially reducing the mixing between the surface freshwater layer and the denser saline layer beneath.

By eliminating trends, we utilized RF models to identify the primary characteristics of the interannual fluctuations of OHC and FWC. The ensemble mean forecast of RF_OHC(VAR)* (Table 2) effectively captures these interannual changes (Fig. 4a), evidenced by a correlation coefficient of 0.9012 and a RMSD of 0.3432. Factors such as 2-meter temperature, wind stress, and evaporation significantly influence the interannual variability of OHC (Fig. 4c). Additionally, total cloud cover and solar radiation have a minor impact on the shape of OHC.

In the RF_FWC(VAR)* model, we incorporated bottom salinity from the Bornholm Basin as a supplementary feature. The direct calculation of salt transport from model data across a section at the Baltic Sea entrance is error-prone. Utilizing daily average cross-section velocities and salinities overlooks high-frequency fluctuations with considerable residual salt flux. The model's precision in predicting accurate salinity levels at the Baltic Sea's entrance is quite low (Lindenthal et al., 2024). Time series of bottom salinity changes in the Arkona and Bornholm Basins facilitate the tracking of the intermittent nature of water inflow and outflow events. The Arkona Basin, being relatively shallow, is known for its dynamic nature regarding volume and salt transport. Here, bottom salinity reflects the salinity shifts caused by inflow and outflow variations at the Baltic Sea entrance. These variations mask the large volume inflows chiefly responsible for the Baltic Sea's salt influx, thus not significantly affecting the Arkona Basin's bottom salinity over time. Conversely, the Bornholm Basin's greater depth means its bottom salinity is less affected by the upper layer's varying salinity water movements. Hence, the Bornholm Basin's bottom salinity serves as a more accurate indicator of the Baltic Sea's salt inflow. We also factored in the annual average river runoff (product ref. no. 3) into the Baltic Sea in our RF model.

The ensemble mean predictions of the RF_FWC(VAR)* are marginally less precise, with a correlation coefficient of 0.8994 and a root mean square difference of 0.3624. The bottom salinity in the Bornholm Basin—used here as an indicator of salt flux into the Baltic Sea—along with total precipitation and the zonal wind component, emerge as the primary drivers of interannual variations in freshwater content (FWC) (Fig. 4d). In contrast, riverine freshwater discharge shows no significant

impact on FWC variability at the interannual scale. Raudsepp et al. (2023) showed that there are multi-year periods when river runoff is in phase or out of phase with the FWC as calculated for the whole Baltic Sea.

Notable FWC peaks occurred in 1993, 2002, and 2013, each followed by a rapid decline in subsequent years (Fig. 4b). The elevated FWC in 1993 reflects the end of a preceding stagnation period characterized by low salinity, which was interrupted by the Major Baltic Inflow (MBI) of 1993 occurring at the end of that year. The gradual increases in FWC observed from 1997 to 2002 and from 2004 to 2013 represent periods during which the influence of earlier MBIs—specifically those of 1993 and 2002—on the basin's total salinity diminished over time.

Reductions in FWC are associated with increases in water salinity, driven primarily by the advection of saline water through the Danish straits. The highest bottom salinity values correspond to the MBIs that occurred at the end of 1993, 2002, and 2014. These inflows had a limited effect on annual FWC during the years of the inflows themselves (1993 and 2002), with their primary impact becoming evident in the following years—1994 and 2003, respectively. Although the 2014 MBI took place at the end of that year, an increase in deep-water salinity was already underway prior to the event, leading to a decrease in FWC during 2014.

Finally, profiles of salinity, temperature, and dissolved oxygen concentration in the Gotland Basin from 1993 to 2023—sourced from the Copernicus Marine Service Baltic Sea in situ multiyear and near real-time observations (INSITU_BAL_PHYBGCWAV_DISCRETE_MYNRT_013_032) (CMS, 2024c) —complement our analyses of OHC and FWC by providing additional context on the evolution of the Baltic Sea's physical and biogeochemical conditions.

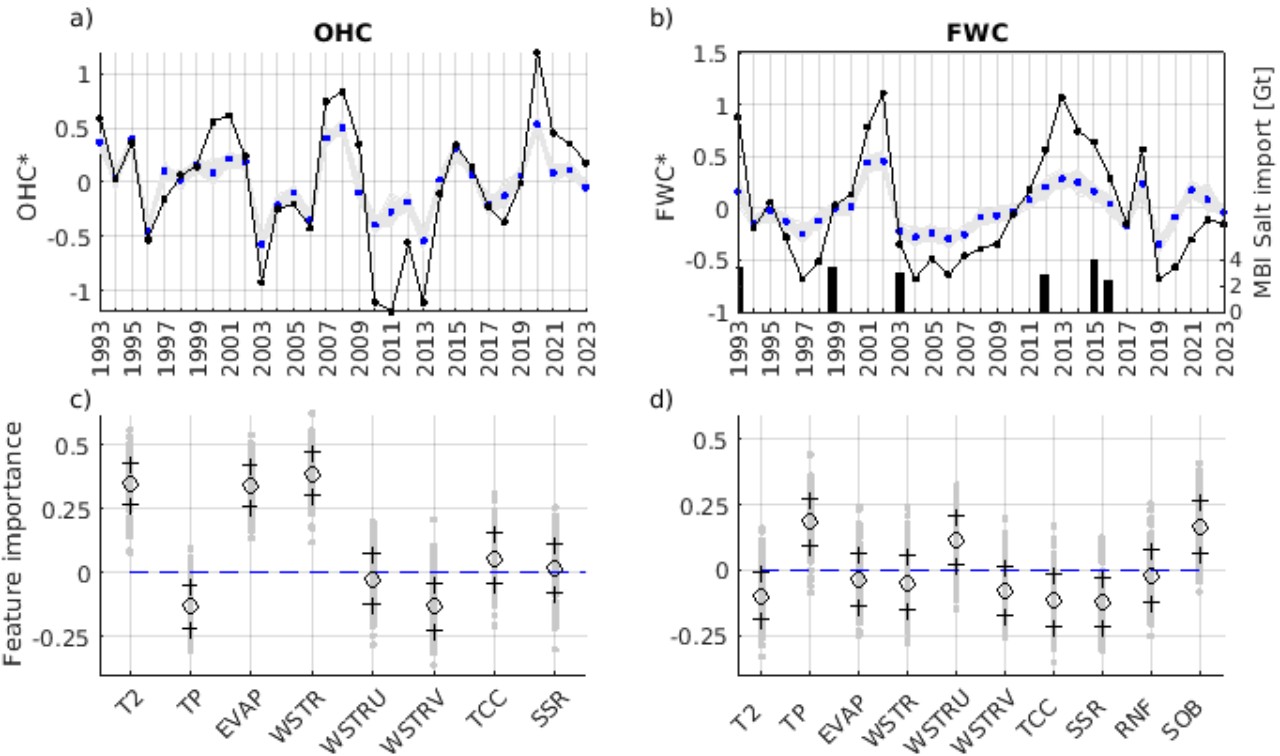

**Figure 4**: Time series of detrended OHC* (a) and FWC* (b) ensemble predictions (ens. mean as blue dots) using RF
ensembles. Ensembles of corresponding models feature importances with ensemble spread ("+" markers corresponding
to 1 STD)  shown on (c) and (d) for OHC and FC respectively. All variables are z-scored. OHC: ocean heat content,
FWC: fresh water content, T2: 2 metre temperature, TP: total precipitation, EVAP: evaporation, WSTR: windstress,
WSTRU and WSTRV: windstress u and v component, TCC: total cloud cover, SSR: surface net solar radiation, RNF:
river runoff, SOB: bottom salinity in the deepest location of the Bornholm basin. Importance values are scaled by the
permutation effect's standard deviation; positive values indicate reduced model performance when a predictor is
permuted, while negative values reflect spurious performance improvements from permutation.

## 4. Discussion and Conclusions

The growing complexity of climate-driven changes in marine environments necessitates a comprehensive framework that
transcends traditional localized assessments. By integrating key indicators into holistic indices representing the overall state
of the sea, this approach advances beyond fragmented analyses to provide a coherent basis for regional evaluation. Such an
integrative methodology is essential for delivering actionable insights that can effectively inform policy and support
sustainable management of ocean resources.

OHC and FWC are established large-scale metrics widely used to track global ocean changes. Here we adapt these metrics to
the regional Baltic Sea and integrate them with additional analysis layers. This framework distinguishes itself by linking
these integral metrics with depth-resolved information and machine-learning-based attribution, which to our knowledge has

not been previously applied in the Baltic Sea context. OHC and FWC are proposed as key descriptors of the Baltic Sea's physical state because they encapsulate the overall thermal and haline content of the entire basin. While temperature and salinity at specific locations or layers provide detailed information, OHC and FWC offer a high-level integration of those details. OHC and FWC reflect temperature and salinity changes across the entire basin. OHC variations primarily follow surface layer temperature changes. The negative trend and interannual variability in FWC are mainly driven by subsurface salinity changes, as surface salinity remains relatively stable (Fig 3c,d). High feature importance values indicate the depths where temperature and salinity changes most closely align with OHC and FWC variations, respectively.

We employed the RF model (Breiman, 2001) to link the atmospheric and hydrologic variables with the variability of OHC and FWC. Given the limited sample size of 31 annual observations, overfitting represents a potential concern in our modeling approach. To mitigate this, we employed an ensemble of 150 independently trained RF models, each with controlled tree complexity (e.g., limited depth, minimum leaf size). This ensemble strategy helps stabilize feature importance estimates and reduces prediction variance arising from random sampling effects, thereby enhancing the robustness of the results. Nonetheless, caution is warranted, as some predictor importances may reflect spurious correlations. Because our RF models were trained on the full time series (1993–2023) with no independent test period, the reported errors (based on OOB) could underestimate true predictive error. The results should thus be interpreted as patterns learned from the given dataset rather than as fully generalizable predictions. Future analyses could leverage extended reanalysis or model datasets (e.g., BMIP; Gröger et al., 2022) to independently validate the machine learning results, thereby strengthening confidence in the predictive skill of the proposed framework.

OHC and FWC are particularly useful for monitoring long-term trends and basin-wide changes, which is why we argue that they effectively define the large-scale physical state. Indeed, our framework's indicators, total OHC and FWC of the Baltic Sea, are integrative and require comprehensive observation or modeling efforts to compute in real-time. In situ monitoring of the entire water column at sufficient spatial coverage is needed to directly measure OHC/FWC, which is more demanding than, say, monitoring a few atmospheric indices. However, these integrated indices provide a succinct summary of the state that individual predictors cannot fully capture. Advancements in remote sensing can help estimate these indices indirectly (e.g. Kondeti and Palanisamy, 2025).

Our results confirm a long-term warming and salinization trend in the Baltic Sea, as evidenced by increasing OHC and a slight decreasing trend in FWC (Table 3). At the same time, by removing these trends for the RF analysis, we isolated the interannual variability and identified its drivers.

Our analysis across the entire Baltic Sea reveals the direct impact of atmospheric forcing on ocean warming. Moreover, this framework provides new insights into the role of salt import/export in FWC's interannual variability, and draws on the basin-wide decline of FWC, elevating the potential role of a flatting MLD from long-term sensible flux change at the air-sea

interface. Particularly, results reveal that the Baltic Sea has undergone substantial change over the past decade as evidenced by the increase in OHC over the last thirty years.

Simultaneously, there has been a reduction in FWC, suggesting an increase in seawater salinity. The analysis of average subsurface temperature and salinity indicates that interannual variations in OHC and FWC are mainly influenced by temperature shifts in both the seasonal thermocline and permanent halocline and changes in salinity within the permanent halocline. This highlights the critical need for a comprehensive framework while reporting on the state of the Baltic Sea, allowing for the evaluation of basin-wide conditions, including its trends, interannual variations, and extremes, as well as the factors driving these changes. Using this approach could prove to be a valuable asset for the science-policy interface, aiding in regional evaluations of the sea state.

Previous studies have reported a positive trend in OHC and a negative trend in FWC (Raudsepp et al., 2022; 2023), along with an inverse relationship between OHC and the maximum ice extent in the Baltic Sea (Raudsepp et al., 2022). The increase in OHC has been attributed to the rising air temperature over the Baltic Sea, yet the decline in FWC remains largely unexplained. Raudsepp et al. (2023) noted that neither salt transport to the Baltic Sea, net precipitation, nor total river runoff accounted for the FWC's downward trend. Despite this, deepwater salinity in the central Baltic Sea has been increasing at a rate of 0.2–0.25 g kg$^{-1}$ per decade (Lehmann et al., 2022). A basin-wide analysis linking FWC changes to atmospheric forces revealed a relation with air temperature, a connection that is physically tenuous, prompting further investigation into other factors. This led to the hypothesis that the decreasing trend in the upper mixed layer thickness in the Baltic Sea might be influencing FWC changes. Over the last three decades, there has been a noticeable reduction in the upper mixed layer depth. While it is plausible to suggest a dynamic relationship between the shrinking mixed layer depth and the decrease in FWC, verifying this hypothesis requires more research than what is covered in the present study.

Interannual variations of OHC are influenced by air temperature, evaporation, and wind stress magnitude over the Baltic Sea (Fig. 4). When considering the lesser impact of total cloud cover and surface net solar radiation, it becomes clear that air-sea heat exchange primarily drives OHC changes in the Baltic Sea. Notably, the annual mean OHC parallels the long-term trend of winter OHC in the Baltic Sea's upper 50-m layer and yearly maximum sea ice extent of the Baltic Sea (Raudsepp et al., 2022), highlighting the coherence of seasonal ice cover and OHC fluctuations. In seas with seasonal ice cover, the characteristics of sea ice are crucial for determining the sea's physical state. Typically, the maximum sea ice extent in the Baltic Sea indicates the severity of the winters (Uotila et al., 2015). Sea ice is vital for temporarily storing ocean heat and freshwater, then releasing it back into the sea (Raudsepp et al., 2022).

The interannual variations of FWC were associated with Major Baltic Inflows, overall precipitation, and zonal wind stress (Fig. 4 d)). The signals of the MBIs are evident in the bottom salinity of the Bornholm Basin. Fig. 4 d) illustrates that interannual variations in FWC are linked to the bottom salinity in the Bornholm Basin, which serves as a proxy for MBIs, as well as zonal wind stress and net precipitation. Therefore, Fig. 4 d) highlights the drivers of FWC, while Fig. 3 d)

emphasizes the significance of halocline salinity's response to FWC. Consequently, we can infer that inflows from the North Sea and net precipitation are responsible for changes in halocline salinity. Because MBIs are short-lived, our use of annual mean wind is a coarse indicator. A high annual mean westerly wind might reflect a generally stormy winter with possible inflows, but it will likely miss isolated inflow events that occur even in otherwise average years. Therefore, we interpret the RF finding of 'zonal wind' importance (Fig. 4d) cautiously – it may be serving as a proxy for the cumulative effect of many small inflows or sustained minor exchange rather than any single MBI. Meier and Kauker (2003) demonstrated that increasing westerly winds could hinder the outflow of freshwater from the Baltic Sea, leading to decreased salt transport into the sea. However, we were unable to directly associate moderate and small inflows from the North Sea with changes in halocline salinity. This aspect requires further investigation and precise simulation of salt transport between the North Sea and the Baltic Sea, which is beyond the scope of the current study. While several studies have underscored a correlation of the Baltic Sea's salinity with river runoff (Kniebusch et al., 2019b; Radtke et al., 2020; Lehmann et al., 2022), our research did not find this connection.

The OHC displays quasi-periodic fluctuations with a period of approximately 5–7 years, with 2020 and 2011 standing out as relative high and low points, respectively (Fig. 4). The elevated wintertime OHC in 2020 coincided with an unusually warm January–March period over the Northern Hemisphere (Schubert et al., 2022), and was accompanied by an exceptionally high marine heatwave index and a large number of marine heatwave days in the Baltic Sea (Bashiri et al., 2024; Lindenthal et al., 2024). In contrast, 2011 featured the most extensive sea ice cover and volume recorded in the past three decades (Raudsepp et al., 2022). Similarly, certain peaks in FWC, such as those observed in 2002 and 2013, align temporally with the years preceding Major Baltic Inflows, while declines in FWC, as seen in 1997 and 2019, occurred following such events. While these specific years are highlighted as examples, they are not the basis for broader conclusions but serve to illustrate patterns consistent with previous studies.

Global warming, with its increased frequency and intensity of extreme events, has had widespread negative impacts on nature and significant socioeconomic repercussions (IPCC, 2021). Our methodology has highlighted the extremes of interannual variability in OHC and FWC. In our study, we utilized the RF model to investigate the relationships between changes in OHC and FWC and their potential drivers. Although the model pinpointed the primary factors, it failed to capture the extremes (Gnecco et al., 2024), as illustrated in Fig. 4a,b. RF models tend to underperform when extreme values are not well-represented in the training data, a common issue in ecological modeling and other practical applications (Fox et al., 2017). This can result in a bias where the model does not recognize or accurately predict rare but impactful events, such as extreme weather conditions, uncommon species occurrences, or anomalies in financial markets (Fox et al., 2017). Acknowledging this, we hypothesize that while primary forces set the stage for extreme events, these events themselves fall outside the scope of standard interannual variability and stem from a distinct combination of forces. Consequently, it is advantageous to analyze extreme events independently from typical interannual variations (Nontapa et al., 2020; Chen et al., 2021). To account for the variations in OHC and FWC, models other than RF, such as deep machine learning models, could

be employed, especially if the temporal resolution is monthly (e.g., Barzandeh et al., 2024) or finer, ensuring a representative dataset is available. It should be noted that the Random Forest analysis reveals statistical connections rather than definitive physical causation. We interpret these connections in light of known mechanisms to ensure they are plausible. Advancing this methodology will further our comprehension of the causes behind extreme events, thereby improving our predictive abilities.

A sustained decline in the Baltic Sea's FWC, indicating increasing salinity, could alert policymakers to intensified saltwater intrusion or reduced freshwater input, prompting investigation into inflow events or drought conditions. Conversely, an ongoing rise in OHC is a clear signal of warming that can inform climate adaptation strategies. The concept of indicators - such as used in this study for OHC and FC, plays an important role facilitating knowledge transfer at the science and policy interface (von Schuckmann et al., 2020; Evans et al., 2025). Integrated indices, OHC and FWC, could be incorporated into regional climate and environmental assessments (HELCOM, 2023) as part of UNEP regional seas conventions (UNEP, 2024), aiding communication of change to stakeholders. Our framework based on an indicator-based approach yields quantitative indicators (annual OHC, FWC, etc.) that can be tracked over time, much like other environmental indicators, to gauge the Baltic Sea's response to climate variability and change.

This framework could be generalized or applied to other regions or to future data. After defining the region of interest and preprocessing relevant data, the three-stage approach combining (i) analysis of OHC and FWC time series, (ii) examination of their vertical distribution, and (iii) RF analysis of their drivers, could be applied.

**Data Availability**

This study is based on public databases and the references are listed in Table 1.

**Author contribution**

**UR** designed the conceptual framework for this study, interpreted the results, and wrote the initial manuscript. **IM** performed the calculations of OHC and FWC, trained the RF models, and prepared the figures; **IM** also contributed to the manuscript development. **PL** and **KvS** contributed to the design of the framework and the presentation of the results. All authors contributed to writing and revising the manuscript.

**Competing Interests**

The authors declare that they have no conflict of interest.

**Disclaimer**

The Copernicus Marine Service offering is regularly updated to ensure it remains at the forefront of user requirements. In this process, some products may undergo replacement or renaming, leading to the removal of certain product IDs from our catalogue.

If you have any questions or require assistance regarding these modifications, please feel free to reach out to our user support team for further guidance. They will be able to provide you with the necessary information to address your concerns and find suitable alternatives, maintaining our commitment to delivering top-quality services.

**Special issue statement**

The paper belongs to the 9th edition of the Copernicus Marine Service Ocean State Report (OSR 9).

**Acknowledgements**

OpenAI's GPT-4o model was used to assist with drafting and editing sections of the manuscript. All content was reviewed, verified, and approved by the authors.

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

 **Appendix 1**

 We also examined the fit of the trend-included time series and their correspondence with meteorological variables for OHC

 and FWC (Figure A1). The correlation coefficient and RMSD for the OHC model are 0.9537 and 0.4310, respectively; for

 FWC model, they are 0.8897 and 0.5994.

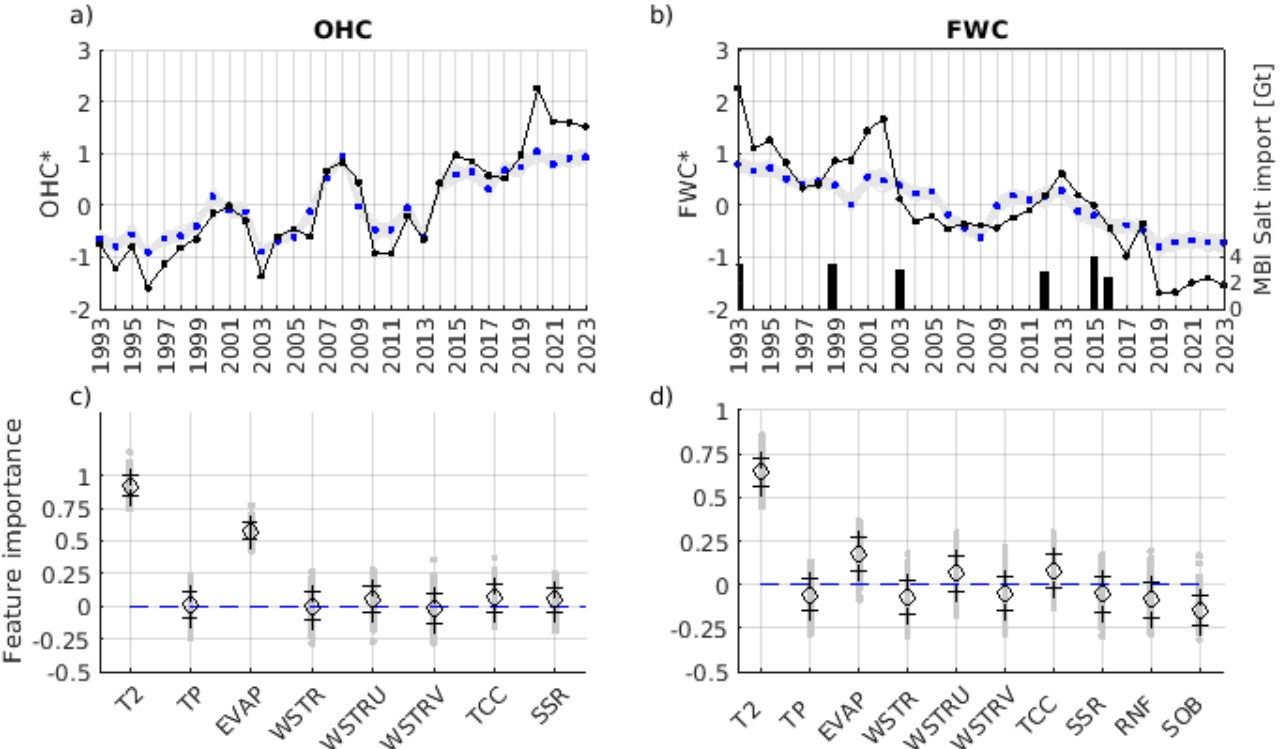

Figure A1. Same as in Figure 4, but the RF  models are fit for the original FWC and OHC including trends.

 **Appendix 2**

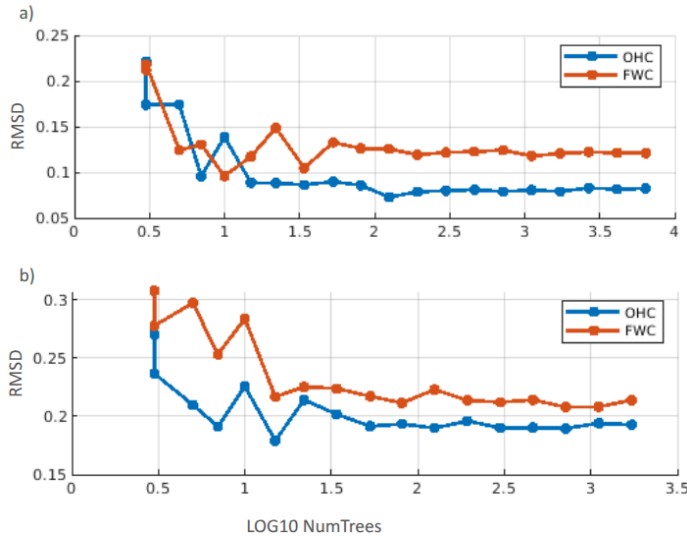

Figure A2. Random forest models for ZAX a) and VAR b) sensitivity to $\log_{10}$ of the number of trees (NumTrees)