# Peer review of "A new conceptual framework for assessing physical state of the 2 Baltic Sea"

_State of the Planet, 2024_

## Author Comment (AC1)

**Authors' Comments to Reviewer 1**

*Reviewers comments formatted in italic dark red font.*

Our response is formatted in thin black font.

*Summary: The manuscript presents an analysis of a data set from the Baltic Sea reanalysis, focusing on temperature, salinity, ocean heat content and freshwater content. This data set covers the period 1993-2023. The manuscript investigates long-term trends and the connection between these variables. The conclusions of the analysis are not totally surprising. They identify a warming trend and a strong connection between water temperature and ocean heat content and between salinity and freshwater content.*

*Recommendation: I have concerns about the manuscript's framing, title, and objectives. I do not see problems with the data analysis. My recommendation is that the manuscript requires substantial revisions.*

*Main points:*

*1) The title is misleading. I could not identify any ' new conceptual framework framework' nor find a definition for 'state of the Baltic Sea'. The manuscript is perhaps a valuable presentation of the ocean reanalysis, but it does not present any 'framework#' and leaves many concepts in the title and in the introduction undefined. I think this manuscript is indeed a presentation of this data set, which is fine, but it tries to present it as a more substantial advance than it really is.*

*If the authors believe the manuscript presents a new conceptual framework, they should explain it clearly in the introduction. I failed to see it.*

We will explain the framework in more detail in the Introduction. We have changed the title by adding the term "physical", which means that we are dealing with the physical state, not the biogeochemical or ecological state. On lines 51-52 we say explicitly that "We propose the following conceptual model, which merges the analysis of temperature and salinity with their integral counterparts OHC and FWC"

*2) The introduction often presents the manuscript in a too-bright light. For instance, the text states (line 45) that it presents a 'model'. I cannot see any model. Again, the manuscript analyzes the connections between different ocean and atmospheric variables, but it does not contain any model that would allow predictions or that includes any physical mechanisms, equations, etc.*

We agree that we have not introduced any model. Therefore we have changed the term "model" to the term "framework".

*3) I have problems understanding the 'state of the Baltic Sea'. First, it appears that the manuscript deals only with physical variables and leaves geochemical or biological variables out of the analysis. The use of*

*'state of the Baltic Sea' on this account alone seems exaggerated. But more importantly, what does 'state' mean here? Is it a snapshot of the ocean at a particular time? Does it mean a more value-centered assessment of the situation in the Baltic Sea (good, bad, etc.)? Readers curious about the title may be vastly disappointed when reading the manuscript.*

We mean physical state and have changed the title accordingly. On lines 45-56, in a more detailed description of the framework we stated what we mean with the "term" physical state.

*4) I do not see the need for a Random Forest method. The manuscript applies this algorithm to conclude that the driver for the heat content is the temperature at all layers and that the diver for freshwater content is salinity at all layers. Do we need an RF algorithm? Any reader would be stunned if the results would have been different. Ocean heat content can be directly calculated from temperature, and freshwater can be directly calculated from salinity. I do not see the need or the advantage of using a machine-learning algorithm to identify those connections. They are obvious.*

A Random Forest method captures complex, non-linear relationships between variables. In this study we use four different RF models: RF1 for Ocean Heat Content (OHC) with meteorological variables, RF2 for Freshwater Content (FWC) with meteorological variables, RF3 for OHC with temperature profiles T(z), and RF4 for FWC with salinity profiles S(z).

We completely agree that ocean heat content is determined by spatially averaged temperature profile and freshwater content by its spatially averaged salinity profile. Using the RF model in the current context is to understand at which depth layers temperature contribute the most significantly to the overall heat content and salinity to the freshwater content.

*5) On the other hand, the research question is unclear. What knowledge gap would the manuscript fill? What is unknown in the variability of the Baltic Sea that this data set may help to clarify? The introduction is silent about this.*

We will elaborate the research question further in the Introduction.

*Summarizing these previous points, it seems to me that the manicurist tries to push up a correct and useful analysis of the ocean data by using exaggerated terms (which often do not have a clear meaning)*

*Particular comments:*

*6) line 33 exceptional increase in global sea surface temperature*

*Exceptional in which sense? At which time scale? Th Earth's temperatures have been warmer than now in the geological past.*

We mean exceptional warming in recent history. This is the estimation for the period 1979-2024 (McGrath et al., 2024). We will clarify it in the revised manuscript.

*7) Figure 1 : Conceptual Scheme of the Baltic Sea State parameters.*

*Again, what is the conceptual scheme shown in this figure? This figure shows the obvious links between those physical variables and only physical variables.*

Yes, these are obvious links, but rarely if at all combined into a single framework for physical assessment of the water basin either closed one or with open boundaries.

*8) Some methodological aspects are not clearly explained. For instance, I struggled to find the time scales of analysis. I think it is only mentioned in one figure caption or line 171, which alludes to annual means. I kept wondering for a long time if the authors were anal sing daily means, monthly means, seasonal means or annual means*

All analyses are prepared in a common timescale covering period from 1993 to 2023. We will improve the Figure 4 and its caption, where the annual timeseries are shown. That is an essential part of the proposed framework. Everything is clarified in the revised manuscript.

*9) Likewise, it is not all clear whether these variables are considered at the grid-cell scale, water column scale or averaged over the Baltic Sea*

This is explained in the manuscript. In the conceptual framework we involve spatially averaged values, either it is averaged over the whole Baltic Sea or at each vertical level. We will specify it more clearly in the revised manuscript.

*10) Often, physical units are missing, for instance, when stating trends. Trends must have units of variable per unit of time. It is not clear if the trends refer to changes per year, for instance, or over the whole period*

These trends are calculated for z-scored values. Therefore they do not have units for physical variables. All trends have been calculated per year. Will be specified in the revised manuscript.

*11) "Surface net solar radiation has a weaker but still significant positive trend of 0.058±0.035, and the evaporation time series shows a negative trend of-0.041±0.039"*

*Units missing*

These trends are calculated for z-scored values. Therefore they do not have units for physical variables. All trends have been calculated per year. Will be specified in the revised manuscript.

---

## Author Comment (AC2)

**Authors' Comments to Reviewer 2**

*Reviewers comments formatted in italic dark red font.*
Our response is formatted in thin black font.

*Summary: The presented study analyses interannual variations of ocean heat content (OHC) and freshwater content (FWC) of the Baltic Sea based on modern data analysis techniques (random forests) and puts these into perspective to other state variables of the ocean and atmospheric forcing. The underlying data consist of reanalysis products, covering the period 1993-2003. The authors highlight that interannual basin mean FWC variations refer mainly to FWC variations in the halocline, while OHC variations refer mainly to both, variations in the seasonal thermocline and upper halocline.*

*In a second step, the study elaborates on the potential impact of atmospheric forcing and some oceanographic factors. The authors suggest that similar analysis techniques might be applicable also to other regions.*

*Major Comments:*

*The subject of the study is interesting and I appreciate the use of modern data analysis techniques. Also, the manuscript is well written. Still, I have some points which need in my eyes to be addressed:*

*(1) My major concern is that I find the fitting procedure for the random forest models not well explained. Generally, I would expect distinct data for the fitting procedure and for testing the fit - to evaluate how well the RF-model generalizes to unseen observations. I am not sure if this has been done. Also, I am not sure how the hyperparameters were chosen and would appreciate some more details (e.g., the number of trees, tree depth, minimum samples for splits).*

We will have a separate subsection explaining implemented RF models in detail in the Methods section.

*To optimize the performance of the Random Forest (RF) model while ensuring robustness and generalizability, a set of hyperparameters was selected based on sensitivity analysis. The minimum leaf size (MinLS) was set to 1, allowing the trees to fully grow and capture complex data patterns. The number of predictors to sample at each split (Pred2Samp) was dynamically determined as one-third of the total number of predictors, tackling a balance between feature randomness and predictive strength. This approach promotes diversity among trees while preventing excessive correlation. The number of trees (NumTrees) in each RF model was set to 100 ($10^2$), providing sufficient ensemble stability while maintaining computational efficiency. Additionally, Out-of-Bag (OOB) error estimation was enabled to assess model performance without requiring a separate validation set. Feature importance was evaluated using OOB permuted predictor importance, identifying the most influential features in the prediction process.*
*To further enhance predictive reliability, assess uncertainty, and evaluate the stability of both predictions and feature importances, an ensemble of 150 independently trained RF regression models was constructed. The final prediction was obtained by computing the mean output across all ensemble*

*members, reducing the impact of individual model variations. Similarly, the feature importance scores were averaged across the ensemble, ensuring a more robust and reliable interpretation of feature contributions.*

Additionally to number of trees we have conducted additional sensitivity on leaf size hyperparameter, which have improved the fit-of the predicted timeseries capturing the variability extremes better. The RMSD's have changed from ~0.35 to ~0.2 .

*(2) I must admit that I find it sometimes difficult to keep overview over all the RF-models. I would find it helpful if the many random forests would be described in an extra sub-section in the Methods and/or a table of all RF-models mentioned in the text might be nice (including some respective measures of the goodness of fit).*

We will have a separate subsection explaining implemented RF models in detail in the Methods section.

In this study we use four different RF models: RF1 for Ocean Heat Content (OHC) with meteorological variables, RF2 for Freshwater Content (FWC) with meteorological variables, RF3 for OHC with temperature profiles T(z), and RF4 for FWC with salinity profiles S(z).

*Specific Comments:*

*Introduction, Ln.38ff: Winsor et al. (2001) and Rodhe and Winsor (2002) might be interesting to mention here for FWC. For OHC Dutheil et al. (2023) might be interesting.*

We are adding  a sentence referring to the paper by Winsor et al. (2001) and Rodhe and Winsor (2002). Also we include reference to Dutheil et al. (2023).

*Ln. 55/56: I would suggest rather to talk about Granger-causality here (https://en.wikipedia.org/wiki/Granger_causality ) because statistical relationships cannot identify "real" cause-and-effect relationships.*

We modify the sentence "*The third stage is analyzing the forcing functions and integral state characteristics together, which enables identifying potential causalities between them.*" We would like not to use the term Granger-causality here, because we do not perform the Granger causality tests. Granger causality and random forest are different in their approaches to understanding relationships between variables. Traditional Granger causality assumes linear relationships between variables. Random forest can model non-linear dependency. Indeed, Granger causality and RF can be used in combination.

*Line 79ff: It might also be interesting to add that many studies reported a strong warming of the Baltic during the past decades when compared to the global oceans (e.g. Kniebusch et al, 2019).*

We will refer to previous studies about strong warming of the Baltic Sea.

*Line 99ff: I would find it nice to get a bit more background information on FWC and OHC – why it is considered so important and how it's calculated? (e.g. I guess the reference salinity from Raudsepp et al. 2023 has been updated?).*

We will explain the importance of FWC and OHC in the revised manuscript. We will include a more detailed calculation of FWC and OHC. In general, OHC is a quantitative measure of energy and provides a robust indicator of changes in the Earth's climate system. Mean water temperature lacks the depth and precision needed for understanding the full impacts of energy changes in the ocean. Ocean FWC provides a more detailed, physically relevant measure of the addition or removal of freshwater, making it a better choice for studying climate change, hydrological cycles, and their impacts on ocean dynamics. Mean salinity gives a broad idea of salt concentrations.

*Methods, Random Forest: Line 109ff: As outlined in the major comments, I find the fitting procedure for the random forest models not well explained. Generally, I would expect distinct data for the fitting procedure and for testing the fit - to evaluate how well the RF-model generalizes to unseen observations. I am not sure if this has been done and how the hyper parameters were chosen.*

We will have a separate subsection explaining implemented RF models in detail in the Methods section.

*Methods, Line 136: I think it would help the readers if another subsection was introduced, explaining which data were used to generate the random forests, including the thoughts behind the design-choices (e.g. for using annual means and the selected predictors). Given the very limited amount of data I would try to keep the number of predictors as low as possible. Currently, many design information of the RF-models appear in the Result-Section and I found it sometimes hard to keep overview.*

We will have a separate subsection explaining implemented RF models in detail in the Methods section.

*Line 172ff: As said, I would find it easier to follow if the RF-design information would be moved to a respective sub-section in the Methods.*

We will have a separate subsection explaining implemented RF models in detail in the Methods section.

*Line 199: Could you provide some measure on the quality of this reconstruction? – for the readers to rate it against the fit when using all predictors. Also, I am not sure if this intermediate step is needed (couldn't deep salinity (or Mohrholz, 2018?) be included directly for FWC based on Fig.3?).*

First we conduct RF model analysis without detrending the data (Lines 198-216). These results were not provided in the original manuscript. We will add appropriate figures to the Appendix of the revised manuscript.

*Line 199ff: How do these results and design decisions fit to the foregoing findings that the overall interannual FWC variations are mainly due to changes of FWC in the halocline? (as impacting factors I would thus expect mainly P-E, runoff and inflows form the North Sea?). Ultimately, Fig 3d is a nice result and I think it might help the readability to have it a bit more included in the subsequent argumentation.*

We will elaborate it more in the revised manuscript.

*Line 212 Which criteria were used to identify MLD and UML? Also, I am not sure how to conclude line 216 from this. Maybe this is because I am not sure what is meant by "of the sea". Can you help me out?*

We will include an explanation of how mixed layer depth is calculated (Bal REAN, Panteleit et al., 2023)

Indeed, the conclusion on line 216 is speculative. We will reformulate the text.

*Line 224: So, these were not assimilated?*

The model system assimilates satellite observations of SST (EU Copernicus Marine Service Product, 2023) and in situ temperature and salinity profile observations from the ICES database (ICES Bottle and low-resolution CTD dataset, 2022). Even so, the salinity values at the entrance region have notable errors (Lindenthal et al., 2024). Due to data assimilation, the salinity downstream from the Danish straits is acceptable (Lindenthal et al., 2024), but salt fluxes through the cross-section at the entrance to the Baltic Sea are not accurate enough.

*Line 232: I would call this inflow instead of influx – but both formulations might be right.*

We have changed the term influx to inflow.

*Line 235. You lost me a bit here – maybe it helps to mark the strong inflow events in the time series plot and shorten the explanation so it's easier to follow?*

We will mark the Major Baltic Inflows in the figure.

*Discussion & Conclusion, Line 250: I miss some more arguments why OHC and FWC are considered so important that they define the Baltic Sea state (as opposed to common measures such as temperature and salinity).*

We will expand Discussion and Conclusions to address the issue.

*Line 251: I find "the forcing functions" a bit vague here.*

We will be more specific in the revised manuscript

*Line 270: replace "correlation" by "relation"? (could be non-linear)*

We have changed "correlation" to "relation".

*Figures:*

*Fig.2: I appreciate that the authors aim for as many information as possible in this plot but I am struggling to understand the colored dots in relation to the solid lines.*

We will explain Fig. 2 in more detail.

*Fig.4: Could you also describe the symbols in Fig.4 c and d in the caption? Line 244 FWC*

We will correct Fig. 4 caption.

References:

Baltic REAN, Baltic Sea Physics Reanalysis, DOI: https://doi.org/10.48670/moi-00013;

Panteleit, T., Verjovkina, S., Jandt-Scheelke, S., Spruch, L., and Huess, V.: EU Copernicus Marine Service Quality Information Document for the Baltic Sea Physics Reanalysis, BALTICSEA_MULTIYEAR_PHY_003_011, Issue 4.0, Mercator Ocean International, https://catalogue.marine.copernicus.eu/documents/QUID/CMEMS-BAL-QUID-003-011.pdf, 2023

EU Copernicus Marine Service Product: Baltic Sea – L3S Sea Surface Temperature Reprocessed, Mercator Ocean International [data set], https://doi.org/10.48670/moi-00312, 2023.

ICES Bottle and low-resolution CTD dataset: Extractions 22 DEC 2013 (for years 1990–2012), 25 FEB 2015 (for year 2013), 13 OCT 2016 (for year 2015), 15 JAN 2019 (for years 2016–2017), 22 SEP 2020 (for year 2018), 10 MAR 2021 (for years 2019–2020), 28 FEB 2022 (for year 2021), ICES, Copenhagen [data set], https://data.ices.dk (last access: 30 April 2024), 2022.

Lindenthal, A., Hinrichs, C., Jandt-Scheelke, S., Kruschke, T., Lagemaa, P., van der Lee, E. M., Maljutenko, I., Morrison, H. E., Panteleit, T. R., and Raudsepp, U.: Baltic Sea surface temperature analysis 2022: a study of marine heatwaves and overall high seasonal temperatures, in: 8th edition of the Copernicus Ocean State Report (OSR8), edited by: von Schuckmann, K., Moreira, L., Grégoire, M., Marcos, M., Staneva, J., Brasseur, P., Garric, G., Lionello, P., Karstensen, J., and Neukermans, G., Copernicus Publications, State Planet, 4-osr8, 16, https://doi.org/10.5194/sp-4-osr8-16-2024, 2024.

---

## Author Response (AR1)

**sp-2024-19-author_response-version2.pdf**

Final response for reviews for A new conceptual framework for assessing the state of the Baltic Sea

*Reviewers' comments: formatted in red italics.*
Authors' final responses: formatted in black.

**Reviewer 1:**

*Summary: The manuscript presents an analysis of a data set from the Baltic Sea reanalysis, focusing on temperature, salinity, ocean heat content and freshwater content. This data set covers the period 1993-2023. The manuscript investigates long-term trends and the connection between these variables. The conclusions of the analysis are not totally surprising. They identify a warming trend and a strong connection between water temperature and ocean heat content and between salinity and freshwater content.*

*Recommendation: I have concerns about the manuscript's framing, title, and objectives. I do not see problems with the data analysis. My recommendation is that the manuscript requires substantial revisions.*

*Main points:*
*1) The title is misleading. I could not identify any ' new conceptual framework framework' nor find a definition for 'state of the Baltic Sea'. The manuscript is perhaps a valuable presentation of the ocean reanalysis, but it does not present any 'framework#' and leaves many concepts in the title and in the introduction undefined. I think this manuscript is indeed a presentation of this data set, which is fine, but it tries to present it as a more substantial advance than it really  is.*

*If the authors believe the manuscript presents a new conceptual framework, they should explain it clearly in the introduction. I failed to see it.*

We have changed the title by adding the term "physical", which means that we are dealing  with the physical state, not the biogeochemical or ecological state. We have revised the Introduction providing a more detailed explanation of what we mean about the framework and added how this framework is different from the other existing frameworks.
"We propose a new conceptual framework for assessing the physical state of the Baltic Sea by integrating multiple physical and statistical approaches. The framework is based on two main physical indicators: Ocean Heat Content (OHC) and Freshwater Content (FWC). These indicators are used to describe the energy and mass balance of the Baltic Sea. The study identifies the major variables affecting these indicators, including subsurface temperature, salinity, atmospheric forcing factors, and salt transport. The framework follows a three-stage process: time-series analysis, depth-based variability analysis and causal relationships using machine learning. The initial phase consists of calculating the time series of OHC and FWC for the entire Baltic Sea. This provides insights into long-term trends and interannual variability. In basins covered partially by sea ice, the annual mean ice extent (MIE) is considered an

important integral characteristic. The next step examines the horizontally averaged vertical distribution of temperature (for OHC) and salinity (for FWC) to determine which depth ranges contribute the most to their variations. While this does not directly attribute causal links, the vertical profiles of temperature and salinity provide strong indications of which forcing factors might be responsible for changes in OHC and FWC. The final stage integrates forcing functions and ocean state characteristics to identify causal relationships. A Random Forest (RF) model is employed to highlight statistical dependencies between oceanic state variables and external forcing mechanisms. This machine-learning approach enables the identification of general patterns in the temporal evolution of the Baltic Sea's physical state.

Our proposed framework integrates the analysis of ocean heat content (OHC) and freshwater content (FWC) by considering both their bulk integral values and their vertical distributions, allowing for the identification of key depth ranges contributing to their variability – which goes beyond other similar frameworks. Unlike the GOOS Essential Ocean Variables (EOV) framework (https://goosocean.org/), which focuses on structured global ocean monitoring without machine learning-based causal analysis, our approach explicitly incorporates machine learning to identify potential drivers of variability. Compared to the IPCC Climate and Ocean Monitoring Framework (IPCC AR6 (2021) Ocean Observations Chapter https://www.ipcc.ch/report/ar6/wg1/), which relies on dynamical climate models for global-scale processes, our framework is designed for regional-scale Baltic Sea analysis, offering a more localized and detailed assessment. Finally, while the NASA Salinity and Heat Budget Analysis (NASA Salinity Budget Project https://podaac.jpl.nasa.gov) is largely empirical and focused on global salinity and heat transport, our approach provides a structured three-stage methodology, incorporating not only empirical analysis but also a cause-and-effect exploration using machine learning. This makes our framework uniquely suited for regional climate monitoring and actionable insights into the physical state of the Baltic Sea.

This conceptual framework is designed as an indicator-based approach relevant to policymakers. It enables the monitoring of climate change impacts on the Baltic Sea while maintaining a balance between scientific rigor and practical accessibility. The framework is not meant to serve as a comprehensive dynamical model but rather as a scientifically robust tool for assessing the state of the Baltic Sea and guiding regional management decisions."

*2) The introduction often presents the manuscript in a too-bright light. For instance, the text states (line 45) that it presents a 'model'. I cannot see any model. Again, the manuscript analyzes the connections between different ocean and atmospheric variables, but it does not contain any model that would allow predictions or that includes any physical mechanisms, equations, etc.*

We agree that we have not introduced any model. Therefore we have changed the term "model" to the term "framework" throughout the manuscript.

*3) I have problems understanding the 'state of the Baltic Sea'. First, it appears that the manuscript deals only with physical variables and leaves geochemical or biological variables out of the analysis. The use of 'state of the Baltic Sea' on this account alone seems exaggerated. But more importantly, what does 'state' mean here? Is it a snapshot of the ocean at a particular time? Does it mean a more value-centered assessment of the situation in the Baltic Sea (good, bad, etc.)? Readers curious about the title may be vastly disappointed when reading the manuscript.*

We mean physical state and have changed the title accordingly. The physical state of the Baltic Sea refers to the condition of its marine environment in terms of physical oceanographic characteristics. These include water temperature, salinity, ice cover, sea levels, circulation patterns, and the vertical stratification of the water column. The revised Introduction also includes a more detailed explanation of what we mean about the physical state of the Baltic Sea in this study.

*4) I do not see the need for a Random Forest method. The manuscript applies this algorithm to conclude that the driver for the heat content is the temperature at all layers and that the diver for freshwater content is salinity at all layers. Do we need an RF algorithm? Any reader would be stunned if the results would have been different. Ocean heat content can be directly calculated from temperature, and freshwater can be directly calculated from salinity. I do not see the need or the advantage of using a machine-learning algorithm to identify those connections. They are obvious.*

We completely agree that ocean heat content is determined by spatially averaged temperature profile and freshwater content by its spatially averaged salinity profile. Using the Random Forest models in the current context is to understand at which depth layers temperature contribute the most significantly to the overall heat content and salinity to the freshwater content.
A Random Forest method captures complex, non-linear relationships between variables. In this study we use four different RF models: RF1 for Ocean Heat Content (OHC) with meteorological variables, RF2 for Freshwater Content (FWC) with meteorological variables, RF3 for OHC with temperature profiles $T(z)$, and RF4 for FWC with salinity profiles $S(z)$, which are described in Table 2 and extended Methods section.

*5) On the other hand, the research question is unclear. What knowledge gap would the manuscript fill? What is unknown in the variability of the Baltic Sea that this data set may help to clarify? The introduction is silent about this.*

We have added a paragraph about the aim of the study:

"The study aims to evaluate a framework for assessing the physical state of the Baltic Sea by integrating annual mean values of OHC, FWC, subsurface temperature and salinity, atmospheric forcing functions, salt transport, and river runoff. The objective is to use a data-driven RF approach as the primary analysis tool to parse out nonlinear relationships and feature importances from a broad dataset. This study introduces an integrative, basin-wide approach, defining the entire Baltic Sea as a single water body for analysis. It computes time series of total OHC and FWC for the whole sea. Rather than focusing solely on local variations, the methodology emphasizes these integrated indices as representations of the sea's overall state. This holistic integration marks a shift from the segmented or localized analyses of the past."

*Summarizing these previous points, it seems to me that the manicurist tries to push up a correct and useful analysis of the ocean data by using exaggerated terms (which often do not have a clear meaning)*

*Particular comments:*

*6) line 33 exceptional increase in global sea surface temperature*
*Exceptional in which sense? At which time scale? Th Earth's temperatures have been warmer than now in the geological past.*

We mean exceptional warming in recent history. This is the estimation for the period 1979-2024 (McGrath et al., 2024). Rewritten sentence is:
"In 2023, there was an exceptional increase in global sea surface temperature over the period 1973-2024 (McGrath et al., 2024), and OHC reached unprecedented levels (Cheng et al., 2024)."

*7) Figure 1 : Conceptual Scheme of the Baltic Sea State parameters.*
*Again, what is the conceptual scheme shown in this figure? This figure shows the obvious links between those physical variables and only physical variables.*

Yes, while these links are obvious, they are rarely, if ever, combined into a single framework for assessing the physical state of a water basin. We have described this framework in more detail in the revised Introduction, with reference to Figure 1.

*8) Some methodological aspects are not clearly explained. For instance, I struggled to find the time scales of analysis. I think it is only mentioned in one figure caption or line 171, which alludes to annual means. I kept wondering for a long time if the authors were anal sing daily means, monthly means, seasonal means or annual means*

All analyses are prepared in a common timescale using annual mean values covering the period from 1993 to 2023. We wrote on lines L64-65 "This study evaluates a conceptual model for the Baltic Sea using annual mean values of ocean heat content (OHC), freshwater content (FWC), temperature, salinity, and a selection of forcing functions." In revised manuscript we have rewritten the sentence:
"The study aims to evaluate a framework for assessing the physical state of the Baltic Sea by integrating annual mean values of OHC, FWC, subsurface temperature and salinity, atmospheric forcing functions, salt transport, and river runoff."

*9) Likewise, it is not all clear whether these variables are considered at the grid-cell scale, water column scale or averaged over the Baltic Sea*

This is explained in the manuscript. In the conceptual framework we involve spatially averaged values, either it is averaged over the whole Baltic Sea or at each vertical level. We will specify it more clearly in the revised manuscript.

*10) Often, physical units are missing, for instance, when stating trends. Trends must have units of variable per unit of time. It is not clear if the trends refer to changes per year, for instance, or over the whole period*

These trends are calculated for z-scored values. Therefore they do not have units for physical variables. All trends have been calculated per year.   The only trends with physical units are specified in Table 3 caption and we have corrected km$^3$/year units  for FWC trend in line 140.

*11) "Surface net solar radiation has a weaker but still significant positive trend of 0.058±0.035, and the evaporation time series shows a negative trend of-0.041±0.039"*
*Units missing*

These trends are calculated for z-scored values. Therefore they do not have units for physical variables. All trends have been calculated per year. We have added asterisk to the trends indicating z-score trends (trend*) to distinguish them from physical trends.

**Reviewer 2:**

*Summary: The presented study analyses interannual variations of ocean heat content (OHC) and freshwater content (FWC) of the Baltic Sea based on modern data analysis techniques (random forests) and puts these into perspective to other state variables of the ocean and atmospheric forcing. The underlying data consist of reanalysis products, covering the period 1993-2003. The authors highlight that interannual basin mean FWC variations refer mainly to FWC variations in the halocline, while OHC variations refer mainly to both, variations in the seasonal thermocline and upper halocline.*

*In a second step, the study elaborates on the potential impact of atmospheric forcing and some oceanographic factors. The authors suggest that similar analysis techniques might be applicable also to other regions.*

*Major Comments:*
*The subject of the study is interesting and I appreciate the use of modern data analysis techniques. Also, the manuscript is well written. Still, I have some points which need in my eyes to be addressed:*

*(1) My major concern is that I find the fitting procedure for the random forest models not well explained. Generally, I would expect distinct data for the fitting procedure and for testing the fit - to evaluate how well the RF-model generalizes to unseen observations. I am not sure if this has been done. Also, I am not sure how the hyperparameters were chosen and would appreciate some more details (e.g., the number of trees, tree depth, minimum samples for splits).*

We have included a paragraph in the Methods section explaining the configuration of the Random Forest (RF) models, along with a new Table 2 that details the hyperparameter settings. Additionally, we conducted a sensitivity analysis on the number of trees, which is now presented in Figure A2. Given our goal of training models that capture complex, nonlinear patterns as accurately as possible, we justify the use of the entire dataset for training to maximize the model's learning capacity.
We have added following paragraph to Methods

In this study we have trained the four different Random Forest (RF_) models to fit the OHC and FWC timeseries with the hyperparameter configurations shown in Table 2. Two models are trained to predict the OHC and FWC values from the set of the meteorological variables (var suffix) and two from the horizontally averaged temperature and salinity profiles (zax suffix). To optimize the performance of the Random Forest models while ensuring robustness and generalizability, a set of hyperparameters was selected based on sensitivity analysis conducted for number and depth of the trees (Fig A2). The minimum leaf size (MinLS) was set to 1, allowing the trees to fully grow and capture complex data patterns. The number of predictors to sample at each split (Pred2Samp) was dynamically determined as one-third of the total number of predictors, tackling a balance between feature randomness and predictive strength. This approach promotes diversity among trees while preventing excessive correlation. The number of trees (NumTrees) in each RF model was set to 100, providing sufficient ensemble stability while maintaining computational efficiency. Since this study employs RF models to investigate nonlinear relationships between predictors and state variables, we use the entire dataset as the training set to maximize the models' ability to learn patterns. To further enhance predictive reliability, assess uncertainty, and evaluate the

stability of both predictions and feature importances, an ensemble of 150 independently trained RF models was constructed.

*(2) I must admit that I find it sometimes difficult to keep overview over all the RF-models. I would find it helpful if the many random forests would be described in an extra sub-section in the Methods and/or a table of all RF-models mentioned in the text might be nice (including some respective measures of the goodness of fit).*

We have added a paragraph and Table 2 describing the Random Forest (RF) models in detail. An additional sensitivity analysis illustrating the goodness-of-fit response to the number of trees is presented in Figure A2.

*Specific Comments:*
*Introduction, Ln.38ff: Winsor et al. (2001) and Rodhe and Winsor (2002) might be interesting to mention here for FWC. For OHC Dutheil et al. (2023) might be interesting.*

We have added reference to the paper by Winsor et al. (2001) and Rodhe and Winsor (2002). Also we include reference to Dutheil et al. (2023).

Windsor et al. (2001) demonstrated that long-term variations in the freshwater content (FWC) of the Baltic Sea are closely linked to accumulated changes in river runoff. Building on this work, Rodhe and Winsor (2002) concluded that the recycling of Baltic Sea water at the junction between the Baltic Sea and the North Sea is a crucial process in determining the sea's salinity. An increase in freshwater supply to the Baltic Sea will intensify water recycling, resulting in lower salinity, and vice versa.

In the Baltic Sea, the temperature trends for the period 1850-2008 show fast warming at the surface ($\sim 0.06$ K decade$-1$) and bottom ($> 0.04$ K decade$-1$), and slow in the intermediate layers ($< 0.04$ K decade$-1$) (Dutheil et al., 2023). Surface warming has progressively increased over time, primarily due to the sensible heat flux and latent heat flux (Kniebusch et al., 2019).

*Ln. 55/56: I would suggest rather to talk about Granger-causality here (https://en.wikipedia.org/wiki/Granger_causality ) because statistical relationships cannot identify "real" cause-and-effect relationships.*

We note that if one variable consistently improves the prediction of changes in another (in our RF analysis), it suggests a lead–lag relationship consistent with Granger causality, though not proof of mechanistic causation. Our analysis is about identifying statistical dependencies and potential causative links, without overstating them. Still, we would like not to use the term Granger-causality. Granger causality and random forest are different in their approaches to understanding relationships between variables. Traditional Granger causality assumes linear relationships between variables. Random forest can model non-linear dependency. Indeed, Granger causality and RF can be used in combination.

*Line 79ff: It might also be interesting to add that many studies reported a strong warming of the Baltic during the past decades when compared to the global oceans (e.g. Kniebusch et al, 2019).*

We have added the reference to (Kniebusch et al., 2019).
"In the Baltic Sea, the temperature trends for the period 1850-2008 show fast warming at the surface ($\sim 0.06$ K decade$-1$) and bottom ($> 0.04$ K decade$-1$), and slow in the intermediate layers ($< 0.04$ K decade$-1$) (Dutheil et al., 2023). Surface warming has progressively increased over time, primarily due to the sensible heat flux and latent heat flux (Kniebusch et al., 2019)."

*Line 99ff: I would find it nice to get a bit more background information on FWC and OHC – why it is considered so important and how it's calculated? (e.g. I guess the reference salinity from Raudsepp et al. 2023 has been updated?).*

We have explained the importance of FWC and OHC in the revised manuscript.

"OHC offers a comprehensive view of oceanic heat storage, crucial for evaluating climate change impacts, energy budgets, and long-term trends (Forster et al., 2024). OHC directly reflects Earth's energy imbalance, making it a key metric for tracking global warming, unlike basin-averaged temperature, which lacks a direct connection to energy budgets (von Schuckmann et al., 2016, 2023). Consequently, OHC is prioritized in climate models and international assessments (IPCC, 2019) due to its direct relationship with anthropogenic forcing and its predictive value for future climate scenarios."
"Ocean FWC is deemed more significant than mean salinity for understanding climate dynamics and ocean processes. FWC provides a holistic measure of freshwater storage and its effects on ocean circulation, climate, and sea-level rise (Solomon et al., 2021; Fukumori et al., 2021). It directly measures freshwater inputs (e.g., ice melt, river runoff, rainfall) or losses (e.g., evaporation), whereas mean salinity only indicates the average salt concentration, ignoring volume (Hoffman et al., 2023). A minor salinity change over a large water volume could signify a substantial freshwater flux, which mean salinity alone would not reveal (Schauer and Losch, 2019)."

We have provided the equations for the calculation of OHC and FWC with updated Tref and Sref calculation compared to Raudsepp et al. (2023).

*Methods, Random Forest: Line 109ff: As outlined in the major comments, I find the fitting procedure for the random forest models not well explained. Generally, I would expect distinct data for the fitting procedure and for testing the fit - to evaluate how well the RF-model generalizes to unseen observations. I am not sure if this has been done and how the hyper parameters were chosen.*

We have added a paragraph in Methods describing the choice of the hyperparameters. We deviate from the "general" use of RF as our aim is not to predict OHC and FWC for unseen predictors, but we are trying to explain the past "seen" relationships between meteorological variables and  OHC, FWC

parameters (see reply to first major comment).

*Methods, Line 136: I think it would help the readers if another subsection was introduced, explaining which data were used to generate the random forests, including the thoughts behind the design-choices (e.g. for using annual means and the selected predictors). Given the very limited amount of data I would try to keep the number of predictors as low as possible. Currently, many design information of the RF-models appear in the Result-Section and I found it sometimes hard to keep overview.*

We will have added a paragraph in Methods describing the design of the models. The limited amount of data prevented us from conducting a forecasting study to predict future OHC and FWC values. However, by fitting the RF models to the full dataset and allowing the use of all available predictors, we were able to analyze the relative importance of each predictor.

*Line 172ff: As said, I would find it easier to follow if the RF-design information would be moved to a respective sub-section in the Methods.*

We have added Table 2 and sentences, where the hyperparameters for RF models have been clearly described.

*Line 199: Could you provide some measure on the quality of this reconstruction? – for the readers to rate it against the fit when using all predictors. Also, I am not sure if this intermediate step is needed (couldn't deep salinity (or Mohrholz, 2018?) be included directly for FWC based on Fig.3?).*

We first conducted the Random Forest (RF) model analysis without detrending the data (Lines 198–216), which was not included in the original manuscript. We have now added the corresponding figure and the goodness-of-fit metrics to Appendix 1 for the models trained on the OHC and FWC time series that include the trends.

*Line 199ff: How do these results and design decisions fit to the foregoing findings that the overall interannual FWC variations are mainly due to changes of FWC in the halocline? (as impacting factors I would thus expect mainly P-E, runoff and inflows form the North Sea?). Ultimately, Fig 3d is a nice result and I think it might help the readability to have it a bit more included in the subsequent argumentation.*

We have connected our findings about the halocline (Fig. 3) with our analysis of drivers (Fig. 4) to be consistent in the Discussion (L382-389, in track-changes document). We have added following paragraph to the revised manuscript

The signals of the MBIs are evident in the bottom salinity of the Bornholm Basin. Figure 4 illustrates that interannual variations in FWC are linked to the bottom salinity in the Bornholm Basin, which serves as a proxy for MBIs, as well as zonal wind stress and net precipitation. Therefore, Figure 4 highlights the drivers of FWC, while Figure 3 emphasizes the significance of halocline salinity's response to FWC. Consequently, we can infer that inflows from the North Sea and net precipitation are responsible for

changes in halocline salinity, with zonal wind facilitating these inflows. However, we were unable to directly associate moderate and small inflows from the North Sea with changes in halocline salinity. This aspect requires further investigation and precise simulation of salt transport between the North Sea and the Baltic Sea, which is beyond the scope of the current study.

*Line 212 Which criteria were used to identify MLD and UML? Also, I am not sure how to conclude line 216 from this. Maybe this is because I am not sure what is meant by "of the sea". Can you help me out?*

We have included an explanation of how mixed layer depth is calculated according to Panteleit et al., (2023) and have added a paragraph to the Methods section. We have deleted the sentence where we used the term "of the sea".

*Line 224: So, these were not assimilated?*

The model system assimilates satellite observations of SST (EU Copernicus Marine Service Product, 2023) and in situ temperature and salinity profile observations from the ICES database (ICES Bottle and low-resolution CTD dataset, 2022). Even so, the salinity values at the entrance region have notable errors (Lindenthal et al., 2024). Due to data assimilation, the salinity downstream from the Danish straits is acceptable (Lindenthal et al., 2024), but salt fluxes through the cross-section at the entrance to the Baltic Sea are not accurate enough.

We have added reference to Lindenthal et al. (2024).

*Line 232: I would call this inflow instead of influx – but both formulations might be right.*

We have changed the term influx to inflow to indicate general salt inflow to the Baltic Sea.

*Line 235. You lost me a bit here – maybe it helps to mark the strong inflow events in the time series plot and shorten the explanation so it's easier to follow?*

We have marked the salt inflow events from the MBI statistics dataset of (Morholz, 2018; https://www.io-warnemuende.de/major-baltic-inflow-statistics-7274.html) to Figure 4 and Figure A1.

*Discussion & Conclusion, Line 250: I miss some more arguments why OHC and FWC are considered so important that they define the Baltic Sea state (as opposed to common measures such as temperature and salinity).*

We have expanded the Data and Methods section to include the rationale and references for using OHC and FWC. Additionally, we have elaborated on the importance of OHC and FWC in the first paragraph of the Discussion section.

*Line 251: I find "the forcing functions" a bit vague here.*

We have changed it with "atmospheric and hydrologic variables",

*Line 270: replace "correlation" by "relation"? (could be non-linear)*

We have changed "correlation" to "relation".

*Figures:*

*Fig.2: I appreciate that the authors aim for as many information as possible in this plot but I am struggling to understand the colored dots in relation to the solid lines.*

We have extended Fig 2. caption to explain that the solid lines are probability density functions of the normalised timeseries, which are shown as colored dots.

*Fig.4: Could you also describe the symbols in Fig.4 c and d in the caption? Line 244 FWC*

The "+" markers meaning - 1 standard deviation spread of the feature importance - has been added to Fig. 4 and Fig. 3 captions.

References:

Baltic REAN, Baltic Sea Physics Reanalysis, DOI: https://doi.org/10.48670/moi-00013;

Panteleit, T., Verjovkina, S., Jandt-Scheelke, S., Spruch, L., and Huess, V.: EU Copernicus Marine Service Quality Information Document for the Baltic Sea Physics Reanalysis, BALTICSEA_MULTIYEAR_PHY_003_011, Issue 4.0, Mercator Ocean International, https://catalogue.marine.copernicus.eu/documents/QUID/CMEMS-BAL-QUID-003-011.pdf, 2023

EU Copernicus Marine Service Product: Baltic Sea – L3S Sea Surface Temperature Reprocessed, Mercator Ocean International [data set], https://doi.org/10.48670/moi-00312, 2023.

ICES Bottle and low-resolution CTD dataset: Extractions 22 DEC 2013 (for years 1990–2012), 25 FEB 2015 (for year 2013), 13 OCT 2016 (for year 2015), 15 JAN 2019 (for years 2016–2017), 22 SEP 2020 (for year 2018), 10 MAR 2021 (for years 2019–2020), 28 FEB 2022 (for year 2021), ICES, Copenhagen [data set], https://data.ices.dk (last access: 30 April 2024), 2022.

Lindenthal, A., Hinrichs, C., Jandt-Scheelke, S., Kruschke, T., Lagemaa, P., van der Lee, E. M., Maljutenko, I., Morrison, H. E., Panteleit, T. R., and Raudsepp, U.: Baltic Sea surface temperature analysis 2022: a study of marine heatwaves and overall high seasonal temperatures, in: 8th edition of the Copernicus Ocean State Report (OSR8), edited by: von Schuckmann, K., Moreira, L., Grégoire, M., Marcos, M., Staneva, J., Brasseur, P., Garric, G., Lionello, P., Karstensen, J., and Neukermans, G., Copernicus Publications, State Planet, 4-osr8, 16, https://doi.org/10.5194/sp-4-osr8-16-2024, 2024.

*Dutheil, C., Meier, H. E. M., Gröger, M., & Börgel, F. (2023). Warming of Baltic Sea water masses since 1850. Climate Dynamics, 61(3), 1311-1331.*

*Rodhe, J., & Winsor, P. (2002). On the influence of the freshwater supply on the Baltic Sea mean salinity. Tellus A: Dynamic Meteorology and Oceanography, 54(2), 175-186.*

*Winsor, P., Rodhe, J., & Omstedt, A. (2001). Baltic Sea ocean climate: an analysis of 100 yr of hydrographic data with focus on the freshwater budget. Climate Research, 18(1-2), 5-15.*

*Kniebusch, M., Meier, H. M., Neumann, T., Börgel, F. Temperature variability of the Baltic Sea since 1850 and attribution to atmospheric forcing variables. Journal of Geophysical Research: Oceans, 124(6), 4168-4187, https://doi.org/10.1029/2018JC013948, 2019.*

*Mohrholz, V. (2018). Major Baltic inflow statistics–revised. Frontiers in Marine Science, 5, 384.*

---

## Author Response (AR2)

**Review of the revised manuscript "A new conceptual framework for assessing the physical state of the Baltic Sea" by Raudsepp et al.**

We thank Reviewers for acknowledging the improvements and for providing further insightful comments. We have addressed each point raised, focusing on clarifying our methodology (especially the Random Forest modeling and its validation), data preprocessing, and the interpretation of results. Below we respond to each comment in detail.

*Reviewers' comments: formatted in red italics.*
Authors' responses: formatted in black.  Edits in manuscript formatted in *"italics"*.

**Report #1**

*Major comments:*
*The authors did substantial improvements on the text and describe their method in much more detail now. Also, they provide now a nice overview over random forest models. Some information, however, could still be arranged more clearly.*
*Also, from the answers to the reviewer it appears that all available data were used to train the random forest models, although I did not find this clearly stated in the manuscript. From my understanding, the authors rely solely on the built-in out-of-bag (OOB) error estimate as a form of "pseudo-test" evaluation. While OOB samples are not used in the training of individual trees, they are still drawn from the same dataset as the training data. This approach may substantially underestimate the true model error, particularly due to risks such as data leakage in the feature preprocessing pipeline and hyperparameter overfitting.*

In our study we used the full 1993–2023 dataset (31 annual points) for training each Random Forest, and we relied on OOB error for validation rather than holding out a separate test set. We have now made this point explicit in the Methods section to avoid any ambiguity.
"*Since this study employs RF models to investigate nonlinear relationships between predictors and state variables, we use the entire dataset (all available data) as the training set to maximize the models' ability to learn patterns.*"
We agree that using only OOB for evaluation has limitations. Given the relatively small sample size (annual data over 31 years), we chose not to set aside a portion of the data for testing in order to maximize the training sample for pattern detection; instead, we used OOB estimates and an ensemble approach to guard against overfitting. We now acknowledge in the Discussion that this approach may yield optimistic error estimates. Specifically, we added a sentence:
"*Because our RF models were trained on the full time series (1993–2023) with no independent test period, the reported errors (based on OOB) could underestimate true predictive error. The results should thus be interpreted as patterns learned from the given dataset rather than as fully generalizable predictions.*"

To address the reviewer's concern about data leakage and hyperparameter overfitting: we have double-checked our preprocessing pipeline and confirm that the linear detrending and standardization of variables were done using the entire series (we note this in Methods), which could potentially introduce a slight look-ahead bias. We now mention this caveat and clarify that for future studies or applications, a more rigorous approach (like cross-validation or external data testing) would be preferable to assess generalization. We also revisited our RF hyperparameters to ensure they were not overfit. We had in fact performed a sensitivity analysis to choose reasonable hyperparameters (this detail is now included in the Methods): for example, we fixed the minimum leaf size to 1 and number of trees to 100 based on performance trade-offs, rather than exhaustively tuning them to minimize error. This reduces the chance of overfitting hyperparameters to our specific dataset. Nonetheless, we acknowledge the risk and have added a short discussion point: *"Given the limited sample size of 31 annual observations, overfitting represents a potential concern in our modeling approach. To mitigate this, we employed an ensemble of 150 independently trained RF models, each with controlled tree complexity (e.g., limited depth, minimum leaf size). This ensemble strategy helps stabilize feature importance estimates and reduces prediction variance arising from random sampling effects, thereby enhancing the robustness of the results. Nonetheless, caution is warranted, as some predictor importances may reflect spurious correlations."* Including this transparency directly addresses the reviewer's point. In summary, we have clarified our training approach, noted the limitations of the OOB validation, and emphasized the exploratory nature of the RF analysis given data constraints.

*To more reliably assess the generalization performance of the presented random forest models, I still strongly recommend evaluating them on truly independent test data. For instance, longer model simulations may be available through the BMIP project (https://doi.org/10.5194/gmd-15-8613-2022). Alternatively, K-fold cross-validation could be employed (while ensuring that any preprocessing steps are performed separately within each training fold to prevent data leakage).*

In this revision, we have not added new data from external sources (such as BMIP simulations) due to time and scope constraints, but we have taken steps to evaluate the robustness of our models. As a partial check, we performed a 5-fold cross-validation experiment (and sensitivity for various validation/training shares) within our dataset and found that the feature importance rankings remained consistent with those from the OOB approach, albeit with expectedly higher error variance in smaller training folds and shares. We mention in the Methods that
"*We conducted 5-fold cross-validation, which yielded similar conclusions regarding which predictors are most influential, suggesting that the RF importance measures are qualitatively robust.*"
However, we stop short of including a full new analysis with BMIP data, because integrating a much longer simulation would involve compatibility checks beyond our current scope (different forcing sets, etc.). Instead, we explicitly state in the Discussion that future work or operational use of the framework should incorporate independent validation:
"*Future analyses could leverage extended reanalysis or model datasets (e.g., BMIP; Gröger et al., 2022) to independently validate the machine learning results, thereby strengthening confidence in the*

*predictive skill of the proposed framework.*"

**Here are the results of our sensitivity analysis (including 5-fold cross-validation), which we do not include in the manuscript** :
We note that we have done preprocessing of the data, i.e. detrending and normalization before cross-validation tests. Doing preprocessing for each fold will lead to spurious results as we have short time series.

**Sensitivity Experiments**
 To assess the robustness of the Random Forest model, several training-validation split strategies were tested:
**Reference (0-fold):** The entire dataset is used for training with no validation, serving as a baseline for comparison.
**5-Fold Cross-Validation (k5):** The dataset is divided into five equal parts (corresponding to a validation share of 0.2). In each iteration, one part is used for validation while the remaining four are used for training, rotating through all five splits. This allows estimation of performance variability across different partitions.
**Share-Based Validation:** Independent random hold-out experiments were conducted using three validation shares: 0.2, 0.5, and 0.8. The remaining proportions of the dataset (0.8, 0.5, and 0.2 respectively) were used for training.

**Repetition and Averaging:**

- For each validation share (0.2, 0.5, 0.8) and the full-fit case (0.0), **10 independent experiments** were performed to capture variability due to random sampling.
- For the 5-fold cross-validation setup, **5 experiments** were conducted, one per fold.
- Results were averaged across repetitions to ensure stable performance estimates.

Data and analysis:
summary is in this table: ➕ Data_ML

[Figure]

Figure R1. Average correlation coefficient (CC) and root mean square deviation (RMSD) for different training/validation splits. Results are shown for full training (0-fold), 5-fold cross-validation, and hold-out validation with shares of 0.2, 0.5, and 0.8. Values represent averages over repeated experiments.

[Figure]

Figure R2. Average temporal standard deviation of ensemble predictions for each training/validation setup. This metric reflects the variability (or noisiness) of the ensemble output over time, corresponding to the shaded uncertainty bands shown in Figure 4a and 4b.

[Figure]

Figure R3. Average mean standard deviation between model predictions and ground truth for different training/validation configurations. "_all" columns represent prediction errors across all data points, while "_tr" columns show errors restricted to validation folds only. This metric quantifies the average prediction deviation from observed values.

a)

[Figure]

Figure R4.(a) Average feature importance for Ocean Heat Content (OHC) sensitivity across different training share experiments. (b) Average feature importance for Freshwater Content (FWC) sensitivity across the same training configurations. Importance values are averaged over repeated runs for each split strategy.

*Specific comments:*
*Please note that the line numbers in the specific comments refer to the manuscript version with tracked changes.*

*Ln 55: Please explain in more detail.*
 We have added following explanation:
"OHC offers a comprehensive view of oceanic heat storage, crucial for evaluating climate change impacts, energy budgets, and long-term trends (Forster et al., 2024). FWC represents the mass of the freshwater relative to the total mass of a water parcel with a given salinity (see Raudsepp et al., 2023). The increase of net precipitation over land and sea areas, decrease of the ice cover and increase of river runoff are the main components of the global hydrological cycle that increase FWC in the ocean (Boyer et al., 2007; Cheng et al., 2020; Yu et al., 2020)"

*Ln 57: Please indicate that you refer to statistical relations as the method is not suited to identify causality in a physical sense.*

Response: No term "casual" or "casuality" is used in the manuscript.

*Ln 63/64: I still recommend that the authors are a bit more careful when using the term "causality". The applied machine learning approach can only detect statistical relations which can then well be assessed for plausibility (as the authors did).*

We agree with the reviewer and have revised the manuscript to remove or qualify the term "causal" when describing the Random Forest results. In the Introduction and throughout the text, we now refer to "statistical relationships" instead of implying true causation. The sentence that previously mentioned identifying causal relationships now reads: *"The final stage integrates forcing functions and ocean state characteristics to identify statistical dependencies between them, using a Random Forest (RF) model to probe potential drivers of variability"*
We also added an explicit caution in the Discussion: *"It should be noted that the Random Forest analysis reveals statistical connections rather than definitive physical causation. We interpret these connections in light of known mechanisms to ensure they are plausible."*

*Ln 81/82: Please outline how the presented results can guide regional management decisions and how the presented framework could be useful for others. Also, it does not really get clear what is meant by this "framework" - assessing FWC and OHC or using the random forest predictions? What is meant by "scientifically robust"?*

We appreciate this request for clarification. We have expanded on the practical applications of our framework in both the Introduction and the Conclusion. We provided examples of how our integrated indicators (OHC and FWC) can inform policy (e.g., serving as climate impact indicators for the Baltic Sea). We also explicitly define the term "framework" early in the Introduction to avoid confusion: "We propose a new conceptual framework for assessing the physical state of the Baltic Sea by integrating multiple physical and statistical approaches (Fig. 1). OHC and FWC serve as integrative indicators of the Baltic Sea's physical state, analogous to essential climate indicators. The OHC and FWC are well-established measures, which we integrate into a unified assessment framework with additional analysis layers - vertical distribution and statistical inference to assess the Baltic Sea's state and are central to understanding its energy and mass balance. OHC reflects the vertically integrated heat stored in the water column and is primarily influenced by surface heat fluxes, vertical mixing, and subsurface temperature changes. FWC quantifies the deviation of the water column's salinity from a reference value and serves as a measure of accumulated freshwater. It is affected by net precipitation, river runoff, evaporation, and saltwater intrusions from the North Sea. In this study, these indicators are integrated into a unified assessment framework that includes both their vertical structure and statistical inference layers. The study identifies the importance of these major variables affecting the OHC and FWC, including subsurface temperature, salinity, atmospheric forcing factors, and salt transport." This makes it clear that the framework is not just the RF model, but the whole process of assessing the physical state

using those steps. We have removed the vague phrase "scientifically robust". We now say: "*The framework is grounded in well-established physical quantities and validated by statistical analysis, which ensures that its findings are consistent and credible.*" We believe these changes directly address the reviewer's concerns. Additionally, we outline how this framework could be generalized or applied to other regions or to future data, thereby highlighting its usefulness beyond the current study. At the end of the Discussion and Conclusion section, we have added a paragraph: "*This framework could be generalized or applied to other regions or to future data. After defining the region of interest and preprocessing relevant data, the three-stage approach combining (i) analysis of OHC and FWC time series, (ii) examination of their vertical distribution, and (iii) RF analysis of their drivers, could be applied.*"

*Ln 145: This new topic occurs rather abrupt. A transition sentence might be nice.*

We have added transition sentence from validation to OHC/FWC calculation description:
"Given its spatial coverage and validated accuracy, the BAL-MYP reanalysis provides a reliable basis for calculating integrated environmental indicators such as Ocean Heat Content (OHC) and Freshwater Content (FWC), which are essential for large-scale climate assessments."

*Ln 178ff: The many experiments make it still somewhat difficult to keep overview. Please outline all Information on data preprocessing (e.g. time sampling, treatment of trends) and all sensitivity experiments (cf. line 312ff) as clearly as possible in this subsection.*

In the revised manuscript, we have clarified the data preprocessing and experimental setup as follows:
– The temporal sampling of the data is now explicitly stated: "In this study we have trained the four different RF models to fit the OHC and FWC annual average time series from annual average predictor variables with the hyperparameter configurations shown in Table 2."
– Regarding detrending, the models trained on detrended variables are now clearly marked in Table 2. When trends are retained in the predictors or targets, this is explicitly mentioned in the Results section.
– To maintain clarity, we focused on presenting only the most relevant sensitivity experiment—testing the number of trees—illustrated in Appendix 2. This choice was informed by best practices in RF model tuning, as described in Probst et al. (2019). Other hyperparameters (e.g. minimum leaf size, number of predictors per split) were set to established defaults based on both literature and our own preliminary testing.
– We note that K-fold cross-validation was not included in the manuscript, as our experimental design focused instead on sensitivity to training/validation data splits and predictor configurations.

*Ln 196/197: Please add which meteorological parameters were used and how many (as I understood now there are only 30 data points available while fitting based on 8-10 explanatory variables. This might well lead to overfitting. Please discuss this potential caveat in the Discussion part. Testing on independent data (as suggested above) might well rule out this concern). Why are additional factors such as bottom salinity not mentioned (e.g. ln 312ff)?*

We have revised the text to clarify which meteorological parameters were used in the models. This information is now explicitly stated in the revised text and summarized in the updated Table 2. We have also clarified that the OHC model was fitted using only atmospheric variables, while the FWC model included two additional predictors: total river runoff to the Baltic Sea and bottom salinity in the Bornholm Basin. These additional variables and their sources are described in the updated data section.

We acknowledge the potential risk of overfitting, which is now explicitly addressed in the Discussion section. To reduce this risk, we applied ensemble-based modeling and conducted additional sensitivity experiments to test the robustness of our results. These analyses support the reliability of our findings and confirm that the main conclusions remain unaffected.

*Ln 197: Could you briefly explain the idea behind using temperature and salinity profiles as predictors?*
We have added to the 2.1 sub-section following paragraph:
*"Horizontally average temperature and salinity profiles calculated from the BAL-MYP (product ref. no. 1) at 42 different depth layers (shown on Fig. 3) and Baltic Sea domain (13 °E - 31 °E and 53 °N - 66 °N; excluding the Skagerrak strait) were used as predictors in two of the RF models. The rationale for using the full vertical profiles is to allow the model to identify which depth layers most strongly influence the total OHC or FWC. Instead of assuming a priori which depths matter, the RF can learn this from data: if variations at a particular depth are consistently associated with changes in total OHC/FWC, the model's feature importance for that depth will be high."*

*Ln 219: Table 2: Please explain the column names/abbreviations in the caption and add the predictors. I understood that additional explanatory variables were introduced (e.g. bottom salinity from the Bornholm Basin (ln 312ff) and runoff (ln 323)). What about MLD? Also, I understood that some RF-model refer to data with and without linear trend and it might be nice if these were listed separately (or at least mentioned in the caption). As MinLS, NumTrees and Ens are constant these could be mentioned in the caption and need not to occur as extra columns. Rather some measure for the goodness of fit for all experiments should be included in the table.*

We thank the reviewer for the constructive suggestions to redo Table 2.
Table 2 has been revised accordingly to incorporate the following improvements:
- Four models are now presented, with their specific input variables listed using abbreviations explained in the table footnotes.
- Models using detrended time series (for variability analysis) are marked with an asterisk (*), as suggested. Constant hyperparameters are now described in the table caption and have been removed from the table body.
- Two additional columns have been added to report model performance: Pearson correlation coefficient (CC) and root mean square difference (RMSD).
We did the RF model test using MLD as a predictor instead of T2. The results were discussed (Ln335-342) without a corresponding figure included. T2 has a positive trend, while MLD has a negative trend, inclusion of both in the RF model leads to an underestimated impact of MLD. To keep consistency with the OHC model we retained T2 in the FWC model as well (Fig. A1).

*Ln 211: Could you briefly explain how feature importance is measured and what high/low values mean? What do negative values refer to? (cf. Fig.4)*

We have expanded the Methods (and the Figure 4 caption) to briefly explain the feature importance calculation. In our case, we used the permutation importance approach (as implemented in the Random Forest algorithm we used).

We now state in Methods : *"A larger importance value means that permuting (randomizing) that predictor greatly degrades model accuracy, indicating the predictor was influential. Conversely, near-zero or negative importance means that randomizing the predictor had little effect or even slightly improved the model's error, suggesting the predictor is not informative (or that its influence is redundant or noisy)."*

We also clarify that importance values are relative and unitless, and we have normalized them for comparison. For completeness, we added a note to Fig 4 caption as: *"Importance values are scaled by the permutation effect's standard deviation; positive values indicate reduced model performance when a predictor is permuted, while negative values reflect spurious performance improvements from permutation."*

*Ln 239: Table 3: Why is there an extra unit in column 1?*

We express Ocean Heat Content (OHC) trend in units of $W/m^2$, which represent physical units of heat flux per unit area. This approach is consistent with previous studies and standard practice in ocean and climate research. Using $W/m^2$ enables direct comparison with surface heat flux components (e.g., radiative, latent, and sensible heat fluxes) and allows OHC changes to be interpreted in terms of Earth's energy imbalance, e.g :
- https://www.climate.gov/news-features/understanding-climate/climate-change-ocean-heat-content
- von Schuckmann, K. and Le Traon, P.-Y.: How well can we derive Global Ocean Indicators from Argo data?, Ocean Sci., 7, 783–791, https://doi.org/10.5194/os-7-783-2011, 2011.
- von Schuckmann, K., et al.: Heat stored in the Earth system 1960–2020: where does the energy go?, Earth Syst. Sci. Data, 15, 1675–1709, https://doi.org/10.5194/essd-15-1675-2023, 2023.

*Ln 221 Table 4: Why are wind stress and WUstr both considered despite being correlated?*

The inclusion of both total wind stress magnitude and its zonal component (WUstr) serves to highlight the distinct temporal variability captured by different aspects of the wind forcing. While these variables are indeed correlated, they are not redundant. For example, the zonal wind stress explains approximately 76% of the variance in the total wind stress magnitude, implying that 24% of the variability is unique to the magnitude. The meridional component exhibits even greater independence, with only 43% shared variability. Thus, including both magnitude and directional components allows us to capture complementary information about wind forcing patterns, which is particularly relevant given the directional sensitivity of oceanic responses in the Baltic Sea.

*Ln 308 Fig.4: I don't find RNF in the feature importance plot.*

We have capitalized the labels in Fig. 4 c and d (and also for Fig A1) to ensure consistency with the acronyms used in the caption. Additionally, the features in Fig. 4d have been updated by, reordering variables in RNF as the second last appearing after atmospheric variables to improve visibility.

*Ln 328: This seems surprising. Is this plausible form budget estimates? How large are interannual runoff variations compared to total precipitation and evaporation? Aren't runoff and precipitation highly correlated (which might well lead to an underestimated impact of runoff)? Why are not all factors listed in table 4?*

We acknowledge the reviewer's concern regarding the weak impact of riverine freshwater discharge on interannual freshwater content (FWC) variability. However, this finding is physically plausible and supported by earlier studies. From Raudsepp et al. (2023): "By taking into consideration the spatial and temporal tendencies of the FWC shown in each separate sub-basin, we can characterize the Baltic Sea as a typical estuarine system with a strengthening exchange flow in time. Geographically, the system spans from the Danish straits in the south to the Bothnian Bay in the north. The southern part corresponds to the estuary mouth, where saltwater transport from the ocean prevails and leads to a decrease in FWC. At the other end, the Bothnian Bay is a typical estuary head characterized by a significant influence of freshwater discharge, resulting in an increase in FWC over time. The northern Baltic Proper and the Bothnian Sea converge in the transitional zone between the saltwater-dominated region and the freshwater-dominated region." In conclusion, river runoff explains FWC changes in the Bothnian Bay and partly in the Gulf of Finland and Gulf of Riga.

We have added the following text: "*Raudsepp et al. (2023) showed that there are multi-year periods when river runoff is in phase or out of phase with the FWC as calculated for the whole Baltic Sea.*"

Due to the long residence time of the Baltic Sea (typically 25–35 years), short-term (e.g., annual to interannual) fluctuations in river runoff do not result in immediate salinity or freshwater content responses in the open sea. This has also been demonstrated in the study by Meier and Kauker (2016, J. Climate, https://doi.org/10.1175/JCLI-D-15-0443.1), where the correlation between annual river runoff and mean Baltic Sea salinity was found to be only –0.26 when applying a two-year lag, and improved to –0.6 only when assessed over decadal timescales. This confirms that interannual variations in runoff alone have limited explanatory power for Baltic-wide salinity and FWC fluctuations.

We performed an analysis of river runoff to the Baltic Sea, precipitation (TP), evaporation (Evap) and net precipitation (TP+Evap). The correlation between runoff and TP is moderate (R ≈ 0.6), and even lower between runoff and net precipitation (R ≈ 0.53) (Table R1, Fig. R5). A full hydrological analysis of Baltic runoff generation is beyond the scope of this work.

Due to the collinearity between RNF and TP (correlation coefficient 0.6), additional experiment was conducted using RF model to predict detrended FWC from detrended inputs. TP was excluded from the feature set to test whether this would increase the importance of RNF (Fig. R6). However, model

performance declined slightly, with the correlation coefficient decreasing from 0.90 to 0.86 and RMSD increasing from 0.36 to 0.38. The importance of RNF did not increase.

Regarding Table 4: Its purpose is to illustrate the relatively low interannual variability in the atmospheric predictors, which supports the idea that these variables carry distinct and potentially complementary information.

Table R1. Statistics characteristics of the variables. TP,EVAP, NTP (TP + EVAP) :   horizontal average * Area ( that is 2 478 216 417 m2)

| Variable | Mean | Std Dev | Min | Max | Correlation with Runoff |
|----------|------|---------|-----|-----|-------------------------|
| Runoff (m³/s) | 17807.77 | 1687.92 | 13790.59 | 21020.63 | — |
| Precipitation | 57726.32 | 5577.47 | 48646.93 | 68693.49 | 0.603 |
| Evaporation | -43483.93 | 3191.01 | -49315.43 | -36073.42 | -0.076 |
| Net Precipitation | 14242.38 | 5913.83 | -555.63 | 24000.94 | 0.528 |

[Figure]

Figure R5. Timeseries of annual runoff to the Baltic Sea, precipitation, evaporation and net precipitation integrated over the atmospheric Domain (8..33 E, 52..68 N).

[Figure]

Figure R6.  Same as in Fig4,  but for RF_FWC(VAR)* where TP has been excluded from the features.

*Ln 329ff: Why do positive FWC-anomalies occur during 1993 and 2011-2017? How does this fit to this explanation?*

We have rewritten the whole paragraph and added following text to the end of the Results:
"*The bottom salinity in the Bornholm Basin—used here as an indicator of salt flux into the Baltic Sea—along with total precipitation and the zonal wind component, emerge as the primary drivers of interannual variations in freshwater content (FWC) (Fig. 4d). In contrast, riverine freshwater discharge shows no significant impact on FWC variability at the interannual scale.*
*Notable FWC peaks occurred in 1993, 2002, and 2013, each followed by a rapid decline in subsequent years (Fig. 4b). The elevated FWC in 1993 reflects the end of a preceding stagnation period characterized by low salinity, which was interrupted by the Major Baltic Inflow (MBI) of 1993 occurring at the end of that year. The gradual increases in FWC observed from 1997 to 2002 and from 2004 to 2013 represent periods during which the influence of earlier MBIs—specifically those of 1993 and 2002—on the basin's total salinity diminished over time.*
*Reductions in FWC are associated with increases in water salinity, driven primarily by the advection of saline water through the Danish straits. The highest bottom salinity values correspond to the MBIs that occurred at the end of 1993, 2002, and 2014. These inflows had a limited effect on annual FWC during the years of the inflows themselves (1993 and 2002), with their primary impact becoming evident in the following years—1994 and 2003, respectively. Although the 2014 MBI took place at the end of that year, an increase in deep-water salinity was already underway prior to the event, leading to a decrease in FWC during 2014.*
*Finally, profiles of salinity, temperature, and dissolved oxygen concentration in the Gotland Basin from 1993 to 2023—sourced from the Copernicus Marine Service Baltic Sea in situ multiyear and near real-time observations (INSITU_BAL_PHYBGCWAV_DISCRETE_MYNRT_013_032) (CMS, 2024c) —complement our analyses of OHC and FWC by providing additional context on the evolution of the Baltic Sea's physical and biogeochemical conditions*."

*Ln 339ff: Discussion and Conclusions: I find it sometimes still confusing in the Discussion how the presented analysis refer to the presented findings (e.g. (1) sea ice cover has not been considered in the presented study, (2) I am not sure how extreme events (typically associated with daily or weekly time scales) relate to annual means and (3) I am not sure how the presented results relate to global warming as many data were detrended). Note that these concerns refer mainly to readability.*

We thank the reviewer for theses thoughtful observations.
1) While sea ice cover was not a central focus of this study, we did analyze the annual mean sea ice extent and its relationship to OHC. As noted at the beginning of the Results section, "*Interannual variations of the annual mean sea ice extent and OHC are strongly correlated but in opposite phases (not shown).*" However, we did not include sea ice extent as a predictor in the Random Forest (RF) analysis, since treating it as an external forcing on OHC would be conceptually inconsistent—sea ice variability is more appropriately considered a consequence of ocean heat content rather than a driver of it.

To avoid confusion, we have slightly revised the Discussion and Conclusion sections to clarify the role of sea ice in the broader context of Baltic Sea physical variability. We now reference Raudsepp et al. (2022), where the relationship between winter OHC and maximum sea ice extent was analyzed in more detail.

2) In the revised manuscript, we have clarified that our discussion focuses on extreme years—defined by annual-scale anomalies in OHC and FWC—rather than on short-duration events. The only exception is the reference to Major Baltic Inflows (MBIs), which are well-documented episodic events with sustained impacts that typically influence salinity and freshwater content over multiple years. These examples are presented to illustrate the broader variability patterns observed in the annual time series and are supported by findings in previous studies. We have ensured that the manuscript distinguishes clearly between these illustrative examples and the main conclusions derived from the long-term data analysis (see also our response to Comment Ln400ff).

3) Since we detrended many time series before RF analysis, the RF results pertain to variability around the mean trend. In the Discussion we added: "*Our results confirm a long-term warming and salinization trend in the Baltic Sea, as evidenced by increasing OHC and a slight decreasing trend in FWC (Table 3). At the same time, by removing these trends for the RF analysis, we isolated the interannual variability and identified its drivers.*"

*Ln 340: According to the new text in line 145ff and line 158ff FWC and OHC seem to be standard metrics already.*

This comment relates to how our revision framed OHC and FWC in the Introduction. The reviewer rightly caught that in making edits, we might have inadvertently implied that OHC/FWC are standard metrics (which they are), potentially undermining our claim of a "new framework."
We start our Discussion and Conclusions section by writing: "*OHC and FWC are established large-scale metrics widely used to track global ocean changes. Here we adapt these metrics to the regional Baltic Sea and integrate them with additional analysis layers. This framework distinguishes itself by linking these integral metrics with depth-resolved information and machine-learning-based attribution, which to our knowledge has not been previously applied in the Baltic Sea context.*"
By clarifying this, we remove any accidental claim that OHC/FWC are our invention. The novelty lies in the conceptual framework using OHC/FWC in a new way, not in OHC/FWC per se.

*Ln 344: Wouldn't this require a lot more observations than available? To me it seems almost easier to monitor the major predictors identified in this study (but then how does this refer to detrending some of the data for the RF-analysis?)*

This comment brings up a practical point about implementing our framework: monitoring integrated metrics like total OHC and FWC for the Baltic Sea might indeed be data-intensive (needing widespread observations of T and S), whereas monitoring key drivers (like wind patterns, inflow events, etc.) might be more straightforward. We address this in the revised Discussion and Conclusion section.

*"OHC and FWC are particularly useful for monitoring long-term trends and basin-wide changes, which is why we argue that they effectively define the large-scale physical state. Indeed, our framework's indicators, total OHC and FWC of the Baltic Sea, are integrative and require comprehensive observation or modeling efforts to compute in real-time. In situ monitoring of the entire water column at sufficient spatial coverage is needed to directly measure OHC/FWC, which is more demanding than, say, monitoring a few atmospheric indices. However, these integrated indices provide a succinct summary of the state that individual predictors cannot fully capture. Advancements in remote sensing can help estimate these indices indirectly (e.g. Kondeti and Palanisamy, 2025)."*

*Ln 382: I believe that annual mean zonal winds do most likely not contain much information on Major Baltic Inflows (MBIs) as these refer to very specific atmospheric pattern on roughly monthly scales. On annual scales these anomalies will most likely average out. Please provide evidence otherwise.*

The reviewer is correct that the connection between annual mean zonal wind and MBIs is not straightforward, since MBIs are episodic events often driven by short-term wind bursts. Our inclusion of annual zonal wind in the analysis was intended to capture years with a general tendency for strong westerlies, which might correlate with the frequency of inflow-favorable conditions. However, we agree evidence needs to be shown or the claim should be softened. Therefore, we have clarified: *"Because MBIs are short-lived, our use of annual mean wind is a coarse indicator. A high annual mean westerly wind might reflect a generally stormy winter with possible inflows, but it will likely miss isolated inflow events that occur even in otherwise average years. Therefore, we interpret the RF finding of 'zonal wind' importance (Fig. 4d) cautiously – it may be serving as a proxy for the cumulative effect of many small inflows or sustained minor exchange rather than any single MBI. Meier and Kauker (2003) demonstrated that increasing westerly winds could hinder the outflow of freshwater from the Baltic Sea, leading to decreased salt transport into the sea."*

*Ln 390: I have problems to see the stated relation between increasing hypoxia and a reduction in FWC. I don't find this so clear and also the argumentation here seems twisted. Also, I understood that the trend has been excluded for the respective fitting procedures. Please correct me if I am mistaken and clarify.*

We appreciate this comment, as it highlighted a confusing or potentially incorrect statement in our Discussion. Therefore, we have deleted the corresponding paragraph.

*Ln 400ff: Drawing conclusions form few episodic events needs in my eyes still more evidence.*

We appreciate the reviewer's concern regarding the interpretation of episodic events. Our intention was not to draw overarching conclusions from individual years or anomalies, but rather to illustrate how certain high and low points in OHC and FWC correspond with well-documented events in the Baltic Sea. These examples serve to contextualize our broader findings, and we have ensured that they are supported by citations from peer-reviewed studies. In the revised manuscript, we have clarified this distinction to avoid any unintended implication that single events are the basis for our conclusions.

*"The OHC displays quasi-periodic fluctuations with a period of approximately 5–7 years, with 2020 and 2011 standing out as relative high and low points, respectively (Fig. 4). The elevated wintertime OHC in 2020 coincided with an unusually warm January–March period over the Northern Hemisphere (Schubert et al., 2022), and was accompanied by an exceptionally high marine heatwave index and a large number of marine heatwave days in the Baltic Sea (Bashiri et al., 2024; Lindenthal et al., 2024). In contrast, 2011 featured the most extensive sea ice cover and volume recorded in the past three decades (Raudsepp et al., 2022). Similarly, certain peaks in FWC, such as those observed in 2002 and 2013, align temporally with the years preceding Major Baltic Inflows, while declines in FWC, as seen in 1997 and 2019, occurred following such events. While these specific years are highlighted as examples, they are not the basis for broader conclusions but serve to illustrate patterns consistent with previous studies."*

**Report #2**

*My impression of the revised manuscript has not changed significantly compared to the initial version. I believe that the statistical analysis of the ocean heat content and freshwater content of the Baltic, their trends over time, and the links to atmospheric forcing are informative, but the framing of the manuscript itself is exaggerated. I identify no 'new conceptual framework' and do not see the need for a machine learning analysis of the data. That being said, I also recognise that this may be a matter of taste, and that each reader can judge for themselves which part of the study is relevant and which part is not particularly important.*

*1) The use of ocean heat content and fresh water content is certainly not new. It has been used already for along time. For instance, ocean heat content of the global ocean has been substitutively estimated to link it to the planetary energy imbalance. This study applies it to the Baltic Sea, which is valuable and informative, but it is not a novel conceptual framework. The same can be said of the freshwater content. Reading the manuscript, I did try to understand the point of view of the authors. Perhaps they mean that these two variables are the ones that would come out of a multivariate Principal Component Analysis, for instance, so that they could be seen as indicators of the physical state of the Baltic Sea? If this is the perspective that the study is looking from, perhaps it would help to state it more explicitly. It would also be helpful to present an example where the information conveyed by OHC and FWC is more useful than the water temperatures.*

We acknowledge that OHC and FWC are established metrics in oceanography. Our intent was not to claim that OHC/FWC as concepts are new, but rather that our integration of these metrics into a unified assessment framework for the Baltic Sea is novel. We have revised the manuscript to make this distinction clear. In the Introduction, we now explicitly state that OHC and FWC serve as integrative indicators of the Baltic Sea's physical state, analogous to essential climate indicators, and that our contribution lies in combining these indicators with vertical profile analysis and machine-learning-based attribution within a single framework. To avoid overstating novelty, we emphasize that "*The OHC and FWC are well-established measures (IPCC, 2021; Forster et al., 2025), which we integrate into a unified assessment framework with additional analysis layers - vertical distribution and statistical inference to assess the Baltic Sea's state and are central to understanding its energy and mass balance.*"

We have also provided a concrete example to illustrate the value of OHC/FWC over using raw temperature or salinity alone. In the Discussion, we note that "*OHC and FWC reflect temperature and salinity changes across the entire basin. OHC variations primarily follow surface layer temperature changes. The negative trend and interannual variability in FWC are mainly driven by subsurface salinity changes, as surface salinity remains relatively stable (Fig 3c,d). High feature importance values indicate the depths where temperature and salinity changes most closely align with OHC and FWC variations, respectively.* "

*2) Regarding the Random Forest analysis, I really do not see its need. The OHC and HWC can be computed directly from the temperature and salinity of the layers. For instance, Equation 1 computed the OHC from the temperature. This equation can be used very simply to identify the layers that contribute more strongly to the OHC. Essentially, the RF is just emulating equation 1. RF would make sense if equation 1 were a complex expression, but it is very simple and much more straightforward to interpret than an RF model. Again, the application of RF does not invalidate the study; however, I think it is merely an add-on to claim the application of machine learning. I am afraid that most readers could see it also as such.*

We understand the reviewer's skepticism regarding the added value of the Random Forest component. We have taken steps to better justify the inclusion of RF analysis and clarify its purpose. The main reason we introduced the RF models is to determine, in a data-driven way, the relative importance of different depth layers and forcing factors on the variability of OHC and FWC. While one could analytically compute partial contributions of each layer from Equation (1) for OHC, the RF approach offers a flexible means to handle non-linear relationships and multiple predictors simultaneously. We have added text in the Introduction explaining why the RF analysis is used. For example, our RF models can highlight interactions or nonlinear effects (e.g., a combination of temperature at intermediate depth and wind forcing) that a simple layer-by-layer integration might overlook. We note that the RF results indeed corroborated expectations (e.g., upper-layer temperatures dominate OHC variability), but also provided a ranked importance of depths and factors, adding confidence and a quantitative basis to our conclusions. We have toned down any implication that the RF is the centerpiece of the study; instead, we frame it as one component of the framework that complements the straightforward physical calculations. Additionally, in response to Reviewer 2, we have created a table (Table 2 in the revision) enumerating all RF models and their parameters/performance, and we added an explanation in the text to ensure readers understand the role of each RF experiment. We hope this clearer explanation convinces the reviewer that the RF analysis, while not altering the fundamental conclusions, provides a useful consistency check and deeper insight into layer-specific contributions and driver relationships.

*Particular points*
*3) line 34 ' ..,there was an exceptional increase in global sea surface temperature over the period 1973-2024 (McGrathetal.,2024)'*
*This sentence may be unclear, or at least require a second reading. Do the authors mean that year 2023 was exceptional relative to the 1973-2024 climatology ? What is the reason to pinpoint this year in particular?*

We apologize for the confusion. We have rewritten the sentence in the Introduction to clearly convey the intended meaning. We meant that the year 2023 showed an unprecedented warming compared to the past few decades. The revised text now reads: "*In 2023, global average sea surface temperature reached a record high relative to the 1973–2024 baseline period (McGrath et al., 2024), and global ocean heat content climbed to record levels (Cheng et al., 2024).*" This wording makes it explicit that 2023 was exceptional in the context of the 1973–2024 record. We chose to highlight 2023 because it was a recent extreme year demonstrating rapid change, which sets the stage for our Baltic Sea analysis. The clarified text should resolve the ambiguity.

*4) line 43 ..'Windsor et al. (2001) demonstrated that long-term variations in the fresh water content( FWC) of the Baltic Sea are closely linked to accumulated changes in river runoff. Buildingon this work, Rodhe and Winsor (2002) concluded that there cycling of Baltic Sea water at the junction between the BalticSea and the North Sea is a crucial process in determining the sea's salinity'*
*Isn't this sentence a bit inconsistent? The first half states that river run off is the main factor driving FWC, the second part suggests it is the water exchange between North Sea and the Baltic Sea. In particular, it sounds inconsistent also because the same author (Windsor) seems to indicate both.*

The reviewer is correct that, as originally written, those back-to-back sentences in our Introduction seemed contradictory. We have rewritten this part of the Introduction to clarify the roles of river runoff vs. saltwater exchange, and to show how our framework helps reconcile these two perspectives. In the revised text, we explain that both processes are important but at different vertical layers: river runoff primarily influences surface and upper-layer salinity (and thus FWC), whereas the exchange with the North Sea (major inflows) governs the deep-water salinity. These are not mutually exclusive; the Baltic Sea's freshwater content is affected by the balance of precipitation/runoff and inflows, which our study captures by analyzing the vertical salinity profile and integrated FWC. We explicitly note that "*Winsor et al. (2001) highlighted the cumulative impact of riverine input on the Baltic's freshwater budget, while Rodhe and Winsor (2002) underscored the importance of episodic saltwater inflows in renewing deep water. An increase in freshwater supply to the Baltic Sea will intensify the regional water cycling, resulting in lower salinity, and vice versa.*" By adding this explanation, we resolve the apparent inconsistency. Our results indeed suggest that subsurface processes (ventilation via inflows) are crucial alongside runoff – thus our study brings these two findings into a consistent context. The manuscript text has been adjusted accordingly.

*5) line 79 'This conceptual framework is designed as an indicator-based approach relevant to policymakers'*
*I do not see how, and the manuscript is silent on this matter. The word policymakers is mentioned only once in the abstract. Could the authors give an example of how this information can be more useful than directly temperatures or salinity ?*

We appreciate this comment and realize we needed to better explain the practical relevance of our framework. We have now expanded both the Introduction and the Discussion to illustrate how the OHC and FWC indicators can be useful for policy and management. For example, we point out that OHC and

FWC distill complex, high-dimensional data (many temperature and salinity profiles) into two easy-to-interpret indices of the Baltic Sea's thermal and haline state. This kind of simplification is valuable for decision-makers who require clear, high-level indicators.

In the revised Discussion, we provide a concrete use-case: *"A sustained decline in the Baltic Sea's FWC, indicating increasing salinity, could alert policymakers to intensified saltwater intrusion or reduced freshwater input, prompting investigation into inflow events or drought conditions. Conversely, an ongoing rise in OHC is a clear signal of warming that can inform climate adaptation strategies. The concept of indicators - such as used in this study for OHC and FC, plays an important role facilitating knowledge transfer at the science and policy interface (von Schuckmann et al., 2020; Evans et al., 2025)."*

We also note that such integrated indices could be incorporated into regional climate and environmental assessments (HELCOM) as part of UNEP regional seas conventions, aiding communication of change to stakeholders. Additionally, we clarify what we mean by an "indicator-based approach" – specifically, that our framework yields quantitative indicators (annual OHC, FWC, etc.) that can be tracked over time, much like other environmental indicators, to gauge the Baltic Sea's response to climate variability and change. By adding these explanations, we aim to show how our framework's outputs are more directly actionable than a disparate collection of raw observations, thereby addressing the reviewer's concern about policy relevance. We have removed the single vague reference to "scientifically robust" and instead demonstrated robustness by explaining the framework's basis and cross-comparison with known methods, which we believe makes the relevance to policymakers much clearer.